# Fast myosin binding protein C knockout in skeletal muscle alters length-dependent activation and myofilament structure
Anthony L. Hessel [1] ✉, Michel N. Kuehn[1], Seong-Won Han[1], Weikang Ma [2], Thomas C. Irving[2], Brent A. Momb[3], Taejeong Song [4], Sakthivel Sadayappan [4], Wolfgang A. Linke [1] & Bradley M. Palmer [5] ✉

In striated muscle, the sarcomeric protein myosin-binding protein-C (MyBP-C) is bound to the myosin thick filament and is predicted to stabilize myosin heads in a docked position against the thick filament, which limits crossbridge formation. Here, we use the homozygous Mybpc2 knockout (C2$^{-/-}$) mouse line to remove the fast-isoform MyBP-C from fast skeletal muscle and then conduct mechanical functional studies in parallel with small-angle X-ray diffraction to evaluate the myofilament structure. We report that C2$^{-/-}$ fibers present deficits in force production and calcium sensitivity. Structurally, passive C2$^{-/-}$ fibers present altered sarcomere length-independent and -dependent regulation of myosin head conformations, with a shift of myosin heads towards actin. At shorter sarcomere lengths, the thin filament is axially extended in C2$^{-/-}$, which we hypothesize is due to increased numbers of low-level crossbridges. These findings provide testable mechanisms to explain the etiology of debilitating diseases associated with MyBP-C.

The force-generating unit of striated muscle is the sarcomere and is predominately comprised of an interdigitating hexagonal array of thick and thin filaments (Fig. 1a)[1,2]. Active force generation arises from the interaction of myosin heads projecting from the thick filament and actin in the thin filament via so-called crossbridge cycling[3,4]. Separate from thin-filament-based calcium-dependent regulation of crossbridge formation by troponin/tropomyosin[5], the thick filament also plays an independent regulatory role[6–9]. In passive sarcomeres, each of the ~300 myosin heads per thick filament exists in a conformational state on a spectrum between so-called ON and OFF states that affect crossbridge formation during contraction[10,11]. At one end is the OFF state, where the myosin heads are docked in a quasi-helical arrangement around the thick filament backbone and have a reduced propensity to form a crossbridge upon activation. At the other end is the ON state, where the myosin heads are positioned up and away from the thick filament, making it more likely to form a crossbridge upon calcium activation of the thin filament[12]. In skeletal muscle, there is evidence for a sarcomere length (SL)-dependent transition of myosin heads, where increasing SL shifts myosin heads toward the ON state and is thought to be an important mechanical underpinning of length-dependence of calcium sensitivity (length-dependent activation)[7,13–16]. However, myosin heads may

also transition between ON and OFF states by SL-independent mechanisms, leading to a basal transition of myosin heads towards ON or OFF states while maintaining their SL-dependent property[8,10,17,18].

Myosin-binding protein-C (MyBP-C) is a proposed regulator of the myosin head ON/OFF state by stabilizing some myosin heads in the OFF conformation[19–21]. MyBP-C arises at ~43 nm intervals along the thick filament backbone, interacting with up to 108 myosin heads per thick filament[22,23]. Of particular interest, MyBP-C dysfunction is associated with debilitating human myopathies linked to altered force production[24–27]. Skeletal muscle MyBP-C consists of a chain of 10 subdomains (Fig. 1a) with the C'-terminus bound to the thick filament (C8–C10) and the other N'-terminal domains (C1–C7) pointed away from the thick filament, most likely interacting with myosin heads and the thin filament[22,23,29]. Skeletal muscles contain a mix of fast (fMyBP-C) and slow (sMpBP-C) isoforms that may function differently and are not necessarily fiber-type specific[19,30].

Previously, we reported details on a model where we could induce a rapid removal of the C1–C7 domains of fMyBP-C in the fast-twitch dominant psoas muscle, leaving the thick-filament-associated C8–C10 domains intact. From that study, we observed that cleavage led to an SL-independent movement of myosin heads towards the ON state, but the SL-

[1]Institute of Physiology II, University of Muenster, Muenster, Germany. [2]BioCAT, Department of Biology, Illinois Institute of Technology, Chicago, USA. [3]Department of Kinesiology, University of Massachusetts—Amherst, Amherst, MA, USA. [4]Center for Cardiovascular Research, Division of Cardiovascular Health and Disease, Department of Internal Medicine, University of Cincinnati, Cincinnati, OH, USA. [5]Department of Molecular Physiology and Biophysics, University of Vermont, Burlington, VT, USA. ✉e-mail: anthony.hessel@uni-muenster.de; Bradley.palmer@uvm.edu

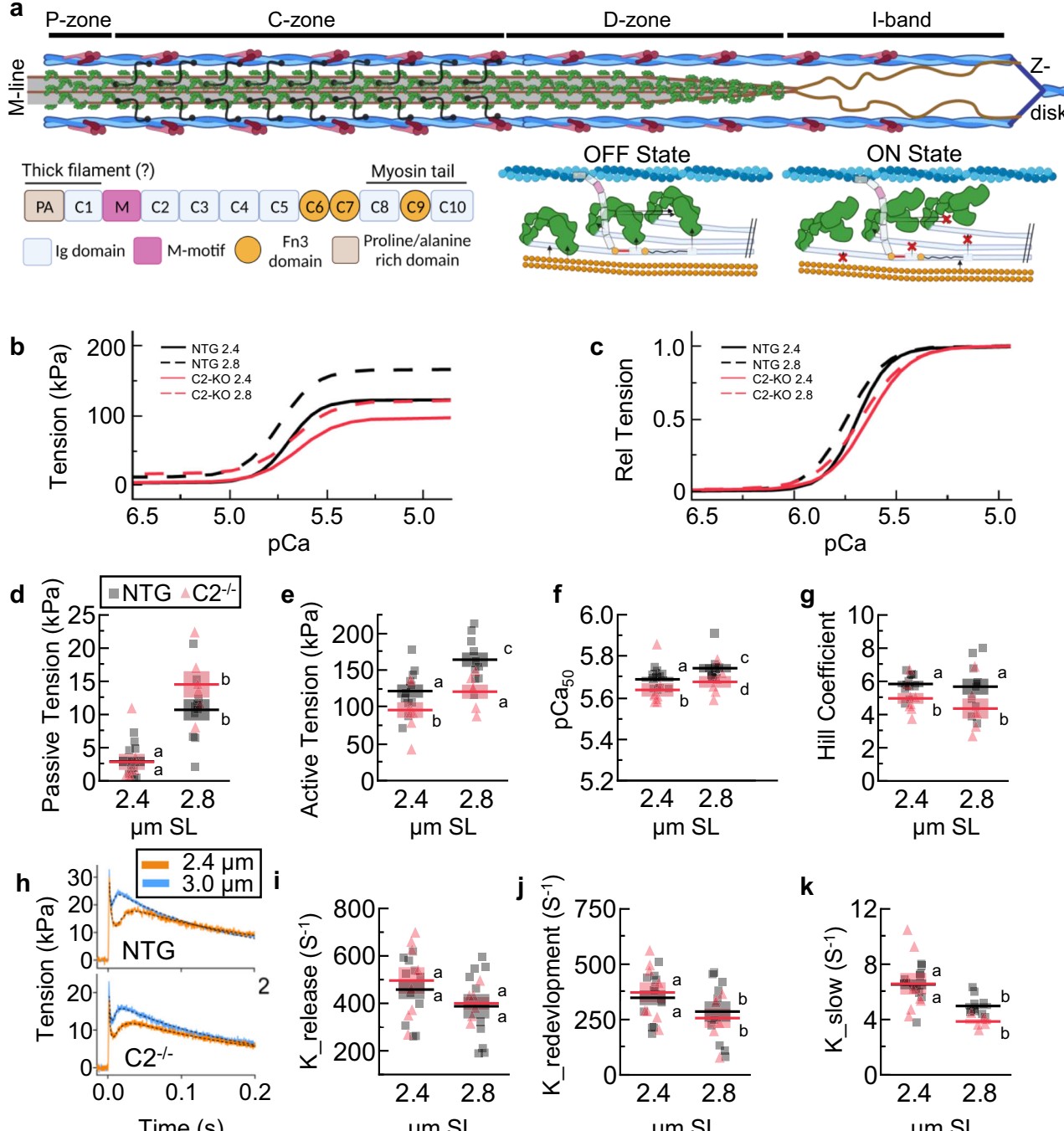

**Fig. 1 | Mechanical assessment of permeabilized C2$^{-/-}$ and NTG fibers from EDL.**
**a** Cartoon representation of a half sarcomere. Thin filaments are comprised of actin filaments (blue) and troponin–tropomyosin complexes (purple). Thick filament backbones (gray) are populated with myosin heads (green), titin filaments (brown), and MyBP-C (black). Thick filaments are demarcated into P-, C-, and D-zones, where MyBP-C is localized in the C-zone. In the I-band, titin extends from the Z-disk to the tips of the thick filament and produces titin-based force as an extensible spring. **b**, **c** Tension-pCa experiments for NTG (black) and C2$^{-/-}$ (red) fiber bundles at 2.4 (solid lines) and 2.8 (dashed lines) μm SL, expressed in absolute terms (**b**) and relative to maximum tension (**c**). Individual data points overlaid on fitted curves are

shown in Supplemental Fig. 1. **d–g** passive tension (**d**), active tension (**e**), pCa$_{50}$ (**f**), and Hill coefficient (**g**) were derived from tension-pCa experiments.
**h** Representative traces of quick-stretch—redevelopment experiments for NTG (top) and C2$^{-/-}$ fibers (bottom) at 2.4 (orange) and 2.8 (blue) μm SL. **i–k** From the quick-stretch—redevelopment experiments, the rate of force release (**i**; K$_{release}$) force redevelopment (**j**; K$_{redevelopment}$) and slow phase (**k**; K$_{slow}$) are calculated. Statistical results are presented as a connecting letters report, where different letters are statistically different (P < 0.05). Data reported as mean ± s.e.m., contain $n$ = 6–11, and evaluated with a mixed-model ANOVA followed by a Tukey's HSD post-hoc analysis. Full statistical details are provided in Table 1.

dependent transition of myosin heads toward the ON state at longer vs. shorter SLs was largely intact[18]. These findings led to the hypothesis that the C1–C7 domains regulate the SL-independent myosin head ON/OFF transitions, while the C8–C10 domains regulate the SL-dependent transitions. If this is correct, then the complete removal of all MyBP-C domains could

ablate, or at least reduce, both SL-independent and SL-dependent controls of myosin ON/OFF states.

To explore the functional role of MyBP-C in skeletal muscle, we studied extensor digitorum longus (EDL) muscles from a fMyBP-C global knockout mouse (C2$^{-/-}$) vs. age-matched non-transgenic (NTG) controls[20]. The NTG

**Table 1 | Mechanical analysis from data in Fig. 1**

| Parameter | Genotype | SL (µm) | N | Mean | s.e.m. | Effect | F | P | Letters |
|---|---|---|---|---|---|---|---|---|---|
| $K_{release}$ (S$^{-1}$) | WT | 2.4 | 11 | 458.09 | 40.12 | Genotype | 0.28 | 0.60 | a |
| $K_{release}$ (S$^{-1}$) | WT | 2.8 | 10 | 387.68 | 51.53 | SL | 2.88 | 0.10 | a |
| $K_{release}$ (S$^{-1}$) | C2$^{-/-}$ | 2.4 | 8 | 496.21 | 56.13 | Interaction | 0.06 | 0.81 | a |
| $K_{release}$ (S$^{-1}$) | C2$^{-/-}$ | 2.8 | 6 | 400.06 | 26.91 | | | | a |
| $K_{red.}$ (S$^{-1}$) | WT | 2.4 | 11 | 348.46 | 33.33 | Genotype | 0.00 | 0.95 | a |
| $K_{red.}$ (S$^{-1}$) | WT | 2.8 | 10 | 286.15 | 46.19 | SL | 4.17 | 0.04* | b |
| $K_{red.}$ (S$^{-1}$) | C2$^{-/-}$ | 2.4 | 8 | 372.55 | 45.85 | Interaction | 0.37 | 0.55 | a |
| $K_{red.}$ (S$^{-1}$) | C2$^{-/-}$ | 2.8 | 6 | 256.54 | 44.46 | | | | b |
| $K_{slow}$ (s$^{-1}$) | WT | 2.4 | 11 | 6.54 | 0.36 | Genotype | 6.23 | 0.02* | a |
| $K_{slow}$ (s$^{-1}$) | WT | 2.8 | 10 | 4.98 | 0.22 | SL | 30.29 | <0.0001* | b |
| $K_{slow}$ (s$^{-1}$) | C2$^{-/-}$ | 2.4 | 8 | 6.58 | 0.78 | Interaction | 3.93 | 0.06 | a |
| $K_{slow}$ (s$^{-1}$) | C2$^{-/-}$ | 2.8 | 6 | 3.86 | 0.18 | | | | b |
| Min. Tension (mN) | WT | 2.4 | 11 | 2.93 | 0.72 | Genotype | 0.70 | 0.41 | a |
| Min. Tension (mN) | WT | 2.8 | 10 | 10.71 | 1.61 | SL | 48.56 | <0.0001* | b |
| Min. Tension (mN) | C2$^{-/-}$ | 2.4 | 8 | 2.86 | 1.21 | Interaction | 1.16 | 0.29 | a |
| Min. Tension (mN) | C2$^{-/-}$ | 2.8 | 6 | 14.54 | 1.99 | | | | b |
| Max. Tension (mN) | WT | 2.4 | 11 | 121.56 | 9.05 | Genotype | 11.28 | 0.002* | a |
| Max. Tension (mN) | WT | 2.8 | 10 | 164.21 | 9.76 | SL | 11.00 | 0.002* | b |
| Max. Tension (mN) | C2$^{-/-}$ | 2.4 | 8 | 95.80 | 10.61 | Interaction | 0.52 | 0.48 | a, c |
| Max. Tension (mN) | C2$^{-/-}$ | 2.8 | 6 | 120.75 | 9.66 | | | | d |
| pCa$_{50}$ | WT | 2.4 | 11 | 5.69 | 0.01 | Genotype | 7.01 | 0.01* | a |
| pCa$_{50}$ | WT | 2.8 | 10 | 5.74 | 0.02 | SL | 4.28 | 0.04* | c |
| pCa$_{50}$ | C2$^{-/-}$ | 2.4 | 8 | 5.64 | 0.03 | Interaction | 0.07 | 0.79 | b |
| pCa$_{50}$ | C2$^{-/-}$ | 2.8 | 6 | 5.68 | 0.03 | | | | d |
| Hill Coeff. | WT | 2.4 | 11 | 5.84 | 0.19 | Genotype | 6.91 | 0.01* | a |
| Hill Coeff. | WT | 2.8 | 10 | 5.68 | 0.48 | SL | 0.58 | 0.45 | a |
| Hill Coeff. | C2$^{-/-}$ | 2.4 | 8 | 4.97 | 0.33 | Interaction | 0.28 | 0.60 | b |
| Hill Coeff. | C2$^{-/-}$ | 2.8 | 6 | 4.36 | 0.61 | | | | b |

The ANOVA analysis F-stats and P-values are provided, as well as a connecting letter report from a Tukey HSD analysis. Data reported as mean ± s.e.m.
*Significant ($P < 0.05$).

EDL is predominately a fast-twitch muscle with a ~ 43% fMyBP-C expression and ~57% sMyBP-C[30]. C2$^{-/-}$ express trace levels of fMyBP-C with a ~15% increase in sMyBP-C that leads to thick filaments with ~33% reduction in total MyBP-C molecules[20]. This model also presents a reduction in myosin heavy chain (Myh4) by 18%, myosin regulatory light chain fast isoform (Mylpf) by 25%, and myomesin (Myom1) by 12%, which collectively suggests a reduced myofilament protein content on the order of 20%[20]. This loss of the force-producing apparatus could contribute to the observed reduction in maximum force production observed in the EDL of these mice[20], but it cannot account for the functional and structural characteristics we found.

Here, we report that compared to NTG, C2$^{-/-}$ fibers present reduced maximum tension and calcium sensitivity but retained length-dependence of calcium sensitivity. C2$^{-/-}$ fibers showed an SL-independent shift toward the ON state, while SL-dependent structural changes in thick and thin filament periodicities were either diminished or not detectable. The underlying mechanisms for this outcome are potentially related to several processes that are not necessarily mutually exclusive, including loss of fMyBP-C, decrease in total MyBP-C content, and/or increased expression of sMyBP-C[20]. Taken together, we provide evidence that MyBP-C plays a role in both the SL-dependent and SL-independent regulation of the myosin ON/OFF level in passive sarcomeres.

## Results and discussion

We first evaluated the mechanical properties of permeabilized fiber bundles from C2$^{-/-}$ and NTG EDL muscles. Tension-pCa measurements were made at SLs of 2.4 and 2.8 µm (Fig. 1b, c). In relaxed fibers (pCa 8), we observed the characteristic increase in passive tension from 2.4 to 2.8 µm SL but no detectable difference between genotypes (Fig. 1d; Table 1). During maximal contraction (pCa 4.5) both NTG and C2$^{-/-}$ had increased tension at the longer SL, but C2$^{-/-}$ produced less active tension across SLs (Fig. 1e; Table 1). NTG fibers increased pCa$_{50}$ at the longer vs. shorter SL (Fig. 1f; Table 1), characteristic of a length-dependent activation[13,14]. While C2$^{-/-}$ fibers also showed an increase in pCa$_{50}$ at the longer SL, values were generally decreased across SLs (Fig. 1f; Table 1), as previously reported at 2.3 µm SL[20], suggesting an SL-independent reduction in calcium sensitivity. No SL effect of the Hill coefficient was detected but there was a significant decrease for C2$^{-/-}$ vs. NTG fibers (Fig. 1g; Table 1), which suggests a general SL-independent impairment to crossbridge recruitment in C2$^{-/-}$ fibers. We next quantified force redevelopment after a small quick stretch that forcibly ruptured crossbridges for a measure of crossbridge kinetics (Fig. 1h). We found no genotype effects for the rate of force release ($K_{Release}$), force redevelopment ($K_{Redevelopment}$), or the slow phase ($K_{slow}$) from the force redevelopment curve (see methods) while there was a significant decrease in these rates at longer vs. shorter SL (Fig. 1i–k; Table 1), as expected[31]. Taken together, C2$^{-/-}$ fibers present deficits in length-dependent enhancements normally observed with force production and calcium sensitivity, but length-dependence of crossbridge kinetics remains largely intact.

We next evaluated myofilament structures using small-angle X-ray diffraction[12]. We collected X-ray diffraction patterns from relaxed (pCa 9) NTG and C2$^{-/-}$ EDL fiber bundles at 2.4 and 2.8 µm SL. The myofilament

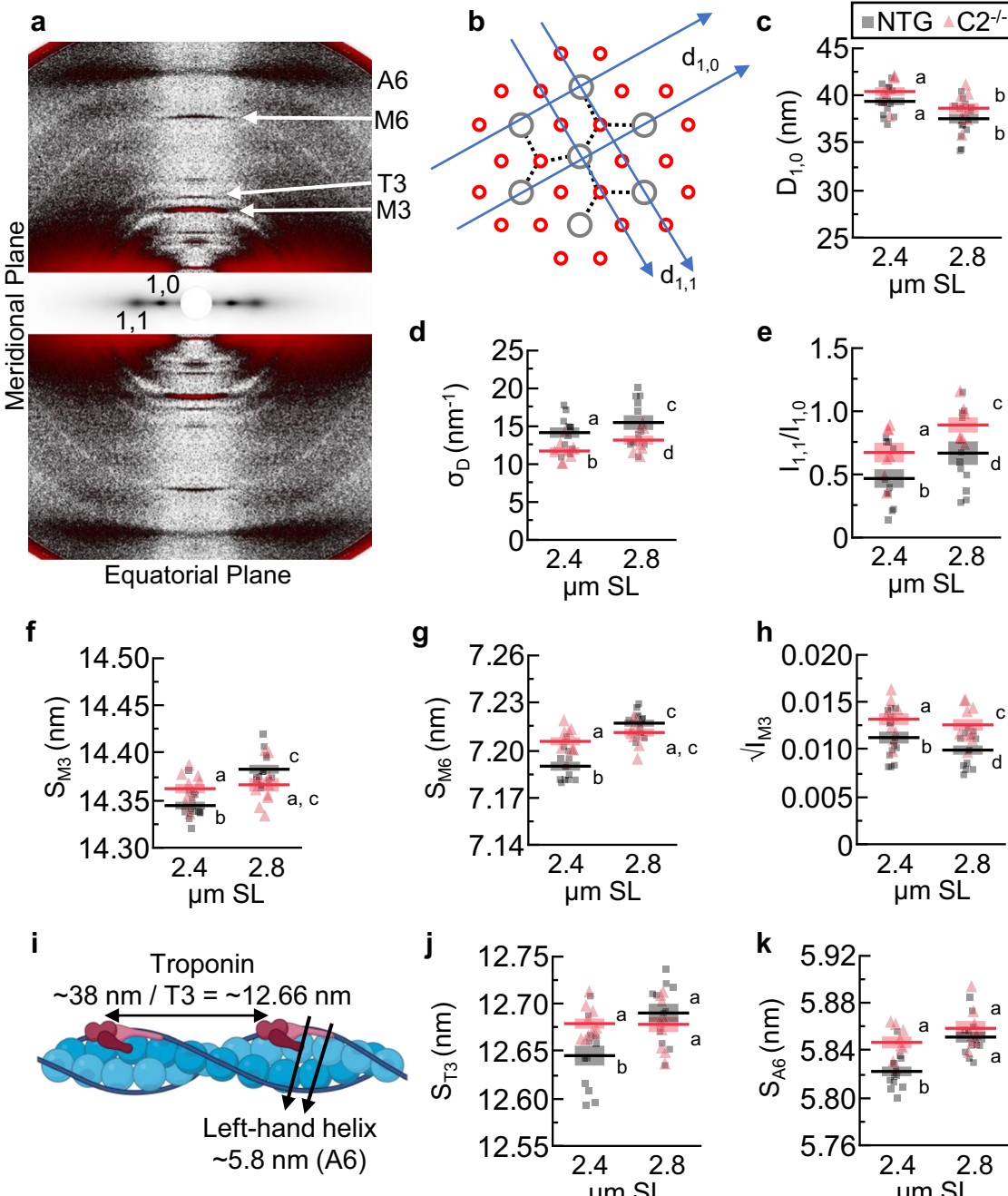

**Fig. 2 | Sarcomere structures of C2$^{-/-}$ (red / triangles) and NTG fibers (black / squares). a** A representative image of an X-ray diffraction pattern, with reflections of interest labeled. The area around the equatorial axis was scaled differently to make reflections easier to view. **b** A cross-section of a myofibril in the thick (gray) and thin (red) filament overlap zone. Example myosin thick-thin filament crossbridges drawn (dotted lines). Overlayed are the geometric lattice planes $d_{1,0}$ and $d_{1,1}$, which lead to the 1,0 and 1,0 equatorial intensities, respectively. **c** $d_{10}$ spacing quantifies lattice spacing. **d** $\sigma_D$ quantifies lattice spacing heterogeneity. **e** $I_{1,1}/I_{1,0}$ is a measure of mass distribution (i.e., myosin heads) between thick and thin filaments. **f** $S_{M3}$ is the periodicity between myosin heads along the thick filament and indicates myosin head orientation. **g** $S_{M6}$ is from a periodicity along the thick filament and quantifies the average thick filament length. **h** $\sqrt{IM3}$ is proportional to the electron density creating the reflection and can be interpreted as quantifying the orderness of myosin heads along the thick filament. **i** A cartoon representation of the thin filament, with periodicities of interest labeled. **j** $S_{T3}$ is the (third-order) axial periodicity of troponin. **k** $S_{A6}$ is the axial periodicity of the left-handed helix of actin and indicates thin filament twisting and elongation. Statistical results are presented as a connecting letters report, where different letters are statistically different ($P < 0.05$). Data reported as mean ± s.e.m., contain $n = 7–14$, and were evaluated with a mixed-model ANOVA followed by a Tukey's HSD post hoc test on significant main effects. Full statistical details are provided in Table 2. Structural data presented as normalized to 2.4 μm SL is shown in Supplemental Fig. 2.

lattice spacing ($d_{1,0}$) was measured by the separation of the 1,0 equatorial reflections (Fig. 2a), which reflects the spacing between 1,0 lattice planes within the filament overlap region (Fig. 2b). $d_{1,0}$ decreased with increasing SL, as expected, with no genotype effect (Fig. 2c; Table 2). Lattice spacing heterogeneity ($\sigma_D$)[32] increased with increasing sarcomere length, as expected[33], but was increased in C2$^{-/-}$ vs. NTG fibers across SLs (Fig. 2a, i;

Table 2). The C2$^{-/-}$ fibers have a relatively complex MyBP-C landscape, with no fMyBP-C and upregulated sMyBP-C that nonetheless leads to a ~ 33% reduction of total MyBP-C content in the mouse EDL. In addition, sMyBP-C and fMyBP-C have well-characterized structural differences in their N-termini that lead to different functionalities that are still not fully described[19,30]. Follow-up studies with sMyBP-C KO and f+sMyBP-C KOs

**Table 2 | Small angle X-ray diffraction analysis from data in Fig. 2**

| Parameter | Genotype | SL (µm) | N | Mean | s.e.m. | Effect | F | P | Letters |
|---|---|---|---|---|---|---|---|---|---|
| $D_{1,0}$ (nm) | WT | 2.4 | 14 | 39.39 | 0.41 | Genotype | 8.28 | 0.13 | a |
| $D_{1,0}$ (nm) | WT | 2.8 | 14 | 37.57 | 0.51 | SL | 97.33 | <0.0001* | b |
| $D_{1,0}$ (nm) | $C2^{-/-}$ | 2.4 | 9 | 40.44 | 0.46 | Interaction | 11.57 | 0.97 | a |
| $D_{1,0}$ (nm) | $C2^{-/-}$ | 2.8 | 9 | 38.68 | 0.54 | | | | b |
| $\sigma_D$ (nm$^{-1}$) | WT | 2.4 | 11 | 14.17 | 0.6866 | Genotype | 5.87 | 0.03* | a |
| $\sigma_D$ (nm$^{-1}$) | WT | 2.8 | 11 | 15.50 | 0.9687 | SL | 22.27 | <0.0001* | c |
| $\sigma_D$ (nm$^{-1}$) | $C2^{-/-}$ | 2.4 | 8 | 11.74 | 0.4896 | Interaction | 0.02 | 0.98 | b |
| $\sigma_D$ (nm$^{-1}$) | $C2^{-/-}$ | 2.8 | 8 | 13.18 | 0.5929 | | | | d |
| $I_{1,1}/I_{1,0}$ | WT | 2.4 | 11 | 0.47 | 0.0735 | Genotype | 11.79 | 0.003* | a |
| $I_{1,1}/I_{1,0}$ | WT | 2.8 | 11 | 0.67 | 0.0931 | SL | 3.54 | 0.04* | c |
| $I_{1,1}/I_{1,0}$ | $C2^{-/-}$ | 2.4 | 7 | 0.76 | 0.0554 | Interaction | 1.13 | 0.34 | b |
| $I_{1,1}/I_{1,0}$ | $C2^{-/-}$ | 2.8 | 7 | 0.89 | 0.0601 | | | | d |
| $S_{M3}$ (nm) | WT | 2.4 | 14 | 14.344 | 0.0041 | Genotype | 5.17 | 0.03* | a |
| $S_{M3}$ (nm) | WT | 2.8 | 13 | 14.383 | 0.0047 | SL | 22.22 | <0.0001* | c |
| $S_{M3}$ (nm) | $C2^{-/-}$ | 2.4 | 10 | 14.362 | 0.0051 | Interaction | 22.52 | <0.0001* | b |
| $S_{M3}$ (nm) | $C2^{-/-}$ | 2.8 | 10 | 14.366 | 0.0071 | | | | a, c |
| $S_{M6}$ (nm) | WT | 2.4 | 12 | 7.190 | 0.003 | Genotype | 0.01 | 0.91 | a |
| $S_{M6}$ (nm) | WT | 2.8 | 12 | 7.217 | 0.002 | SL | 58.77 | <0.0001* | c |
| $S_{M6}$ (nm) | $C2^{-/-}$ | 2.4 | 10 | 7.206 | 0.003 | Interaction | 21.29 | <0.0001* | b |
| $S_{M6}$ (nm) | $C2^{-/-}$ | 2.8 | 10 | 7.211 | 0.003 | | | | a, c |
| $\sqrt{I_{M3}}$ | WT | 2.4 | 14 | 0.0113 | 0.0006 | Genotype | 9.64 | 0.01* | a |
| $\sqrt{I_{M3}}$ | WT | 2.8 | 13 | 0.0099 | 0.0005 | SL | 5.79 | 0.03* | c |
| $\sqrt{I_{M3}}$ | $C2^{-/-}$ | 2.4 | 10 | 0.0132 | 0.0006 | Interaction | 0.86 | 0.36 | b |
| $\sqrt{I_{M3}}$ | $C2^{-/-}$ | 2.8 | 10 | 0.0126 | 0.0006 | | | | d |
| $S_{T3}$ (nm) | WT | 2.4 | 12 | 12.645 | 0.0105 | Genotype | 0.96 | 0.34 | a |
| $S_{T3}$ (nm) | WT | 2.8 | 11 | 12.690 | 0.0098 | SL | 5.79 | 0.01* | c |
| $S_{T3}$ (nm) | $C2^{-/-}$ | 2.4 | 10 | 12.679 | 0.0054 | Interaction | 3.91 | 0.03* | b |
| $S_{T3}$ (nm) | $C2^{-/-}$ | 2.8 | 10 | 12.678 | 0.0082 | | | | d |
| $S_{A6}$ (nm) | WT | 2.4 | 13 | 5.822 | 0.0041 | Genotype | 7.48 | 0.01* | a |
| $S_{A6}$ (nm) | WT | 2.8 | 12 | 5.851 | 0.0047 | SL | 14.20 | <0.0001* | a |
| $S_{A6}$ (nm) | $C2^{-/-}$ | 2.4 | 8 | 5.847 | 0.005 | Interaction | 2.65 | 0.05* | b |
| $S_{A6}$ (nm) | $C2^{-/-}$ | 2.8 | 8 | 5.859 | 0.0062 | | | | a |

The ANOVA analysis F-stats and P-values are provided, as well as a connecting letter report from a Tukey HSD analysis. Data reported as mean ± s.e.m.
*Significant ($P < 0.05$).

would help parse out further details. What can be said currently is that the altered MyBP-C composition impacts lattice order across SLs, suggesting MyBP-C's ability to interact via thin filament interactions, as recently demonstrated[21,22,28].

The distribution of myosin heads between the ON and OFF states is a critical determinant of muscle performance during contraction (Fig. 2e)[8,11,13]. In many mammalian muscles, there is an SL-dependent mechanism where sarcomere stretch extends I-band titin, which increases the titin-based passive tension, resulting in an extension of the thick filament periodicity axially, which is associated with some myosin heads shifting from an OFF towards an ON state[12,33,34]. It is important to note that these ideas are actively evolving. While there is consensus that the spacing of the M6 reflection ($S_{M6}$) reflects the periodicity of structures within the thick filament backbone, whether these small changes in thick filament backbone periodicity (<1%) are due to actual extension of molecules themselves or changes in the molecular packing, or both, has been a subject of considerable debate and is not yet clear[23,35,36]. Similarly, while the OFF to ON transitions seem to be associated with passive stretch in skeletal muscle[11] and in porcine muscle[8], they have not been consistently seen in rodent cardiac muscle under diastolic conditions[9,37]. In any event, the structural changes

responsible for increases $S_{M6}$ appear to disrupt the stabilizing interactions between OFF myosin heads and the thick filament backbone, most likely involving titin and MyBP-C[22,23]. Other mechanisms regulate myosin heads in an SL-independent fashion, such as phosphorylation[38] or the external addition of pharmaceuticals that promote force myosin heads into either OFF or ON states[10,17,33]. Importantly, without SL change, repositioning myosin heads into the ON state is associated with increased $S_{M6}$[10,17,33], most likely due to structural rearrangements within the thick filament backbone that are not yet understood[11]. We tracked three X-ray reflections to study this phenomenon. (1) The intensity ratio between the 1,1 and 1,0 reflections ($I_{11}/I_{10}$), which tracks the radial movement of mass in the form of myosin heads from thick toward neighboring thin filaments[39]. (2) The spacing and intensity of the M3 reflection ($S_{M3}$, $I_{M3}$). $S_{M3}$ represents the average axial periodicity between myosin crowns along the thick filament. $S_{M3}$ does not provide the radial position of the myosin heads off the thick filament backbone but may reflect an orientation change that generally tracks the myosin ON/OFF state. $S_{M3}$ will also change when myosin heads strongly bind to actin, complicating the interpretation[15]. $I_{M3}$ is an indicator of the helical ordering of the myosin heads and is a useful indicator of the degree of ordering of myosin heads. Since the diffracted intensity is proportional to

the square of the total electron density, the square root of $I_{M3}$ ($\sqrt{I_{M3}}$) is directly correlated to the number of diffracting (presumably ordered) myosin heads. All myosin heads can be in different orientations along the thick filament, so increasing or decreasing the homogeneity of myosin head positions will increase or decrease $I_{M3}$, respectively[40]. (3) The spacing of the M6 reflection ($S_{M6}$) reflects changes within the thick filament periodicity length, either due to the extension of individual molecules comprising the backbone or changes in the molecular packing of these molecules within the filament[35].

For NTG fibers, we observed the expected SL-dependent increase of $I_{11}/I_{10}$, $S_{M6}$, $S_{M3}$, and $\sqrt{I_{M3}}$ with increasing SL (Fig. 2e–h; Table 2). Strikingly, $C2^{-/-}$ fibers presented nearly constant values across SLs for $S_{M6}$ and $S_{M3}$. $S_{M6}$ and $S_{M3}$ values in $C2^{-/-}$ fibers were generally elevated compared to the level of NTG fibers at the longer length so that at the short SL, $C2^{-/-}$ values were greater than NTG values (Fig. 2f-g; Table 2). These findings suggest two important conclusions. First, compared to NTG fibers, $C2^{-/-}$ fibers have more myosin heads in the ON position across SLs. This can be caused by destabilization of the OFF state, which seems likely in the C-zone, as OFF-state myosin heads interact with MyBP-C domains $C8^{-1}0^{22,23}$. Second, muscles with diseased, genetically modified, or partially cleaved MyBP-Cs all present evidence of destabilization of at least some C-zone myosin heads in the OFF state[18,20]. MyBP-C appears to position the sub-population of myosin heads in the C-zone into a different orientation than those in the P- or D-zones[40]. In $C2^{-/-}$ fibers, there is a ~ 33% reduction in total MyBP-C molecules per thick filament. All the fast-MyBP-C molecules have been removed, accompanied by a 30% increase in slow MyBP-C that cannot account for all lost fast-isoform molecules. Indeed, $\sqrt{I_{M3}}$ was elevated in $C2^{-/-}$ across SLs (Fig. 2h; Table 2), which indicates more myosin head ordering in the axial direction—something typically associated with myosins transitioning towards the OFF state, but in this case, where other markers indicate the opposite (see above). Increased axial ordering in the presence of more ON state heads in $C2^{-/-}$ could result from the heads showing less angular dispersion in their axial orientations than in WT[41]. What we cannot yet infer from this study is if the removal of fMyBP-C is the sole driver of these changes or if we are observing the effects of a general decrease in the total MyBP-C. Furthermore, sMyBP-C, with its diverse functionality[42,43], may rescue the phenotype if it was upregulated to account for all missing fMyBP-C molecules. sMyBP-C or s+fMyBP-C knockouts would help elucidate this lingering question.

Surprisingly in $C2^{-/-}$ fibers, SL-extension did not modify the thick filament backbone structure or change the axial ordering of myosin heads as is expected from the NTG results but did elevate the average position of myosin heads away from the backbone, as demonstrated by the increased $I_{11}/I_{10}$, albeit with larger values across SL (Fig. 2e–h; Table 2). One possible explanation is that $C2^{-/-}$ fibers were already transitioned to a higher ON state, even at short SL, and so the effect of stretch was reduced (and/or may have occurred below our detection limits). However, we could detect radial head movement from short to long lengths in $C2^{-/-}$ fibers, even though they were naturally transitioned to a more ON state across SLs. This may suggest that the radial movement of myosin heads toward the thin filament ($I_{11}/I_{10}$) is more sensitive to sarcomere length than orientation changes that alter myosin periodicity ($S_{M3}$). It should be noted that the orientation of the blocked myosin head—the counterpart to the free head—also contributes to the M3 reflection and may also alter its typically OFF-state position in $C2^{-/-}$ vs. NTG fibers. The relationship of the M3, M6, and $I_{11}/I_{10}$ in $C2^{-/-}$ could be further studied by forcibly shifting myosin heads into the OFF or ON states using (for example) the myosin deactivator mavacamten[44] or myosin activator dATP[10,45]. Taken together, we find that compared to NTG, the measured alterations of myosin head regulation in the $C2^{-/-}$ can be explained by a partial destabilization of the OFF state.

As a last assessment, we studied thin filament length. By an unknown mechanism, thin filaments backbones change their conformation and/or subtlety elongate with increasing SL in passive mammalian cardiac and skeletal muscle[18,33,37], and we evaluated if this changes in fMyBP-C KO muscle. Evidence from direct visualization experiments[23,28,29] shows so-

called C-links, where the N'-terminal domains of MyBP-C interact with the thin filament, bridging the thick and thin filament. C-links, if present, would theoretically elongate the thin filaments during sarcomere stretch, but since C-links are short compared to an SL change (100's of nm), the N'-terminal would drag along the surface of the thin filament. We quantified thin filament elongation by the spacing of the A6 reflection ($S_{A6}$), which represents the periodicity of the left-handed helix of actin, and the T3 reflection ($S_{T3}$), which represents the $3^{rd}$-order axial spacing of troponin (Fig. 2a, i. In NTG fibers, we observed an increased $S_{A6}$ and $S_{T3}$ in the long vs. short SL (Fig. 2i–k; Table 2). In contrast, $C2^{-/-}$ fibers presented no SL-dependence of $S_{A6}$ or $S_{T3}$ but had longer spacings at the short SL, similar to those with longer SL (Fig. 2i–k; Table 2). While a loss of fMyBP-C and C-links could explain why there is little thin filament extension with increasing SL, it cannot explain why the thin filaments became longer at the short SL in $C2^{-/-}$ vs. NTG fibers. In this study, $C2^{-/-}$ sarcomeres have more ON myosin heads as well as longer thin filaments. In passive muscles, ON myosins likely produce a small number of force-producing crossbridges that generate a small amount of force on the thin filaments[46–49]. These bound crossbridges, estimated at ~2% of myosin heads, increase with increasing proportion of ON myosin heads[50]. In $C2^{-/-}$ fibers, more myosin heads are ON (see above), and so we hypothesize that more force-producing crossbridges are engaged as well, contributing to thin filament extension. This hypothesis can be tested by using mavacamten or dATP to force nearly all myosin heads into the OFF or ON state, respectively[10,44], and tracking changes to thin filament length. It is not known how this increased pool of force-producing myosin crossbridges and changes in thin filament length modulate muscle contraction.

## Methods
### Animal model and muscle preparation
Animal procedures were performed according to the *Guide for the Use and Care of Laboratory Animals* published by the National Institutes of Health and approved by the institutional animal care and use committee at the University of Vermont and the University of Cincinnati. Homozygous $C2^{-/-}$ (FVB strain) mice of either sex and $14^{-16}$ weeks old were deeply anesthetized with 2–4% isoflurane and killed by cervical dislocation. Skeletal muscles including extensor digitalis longus (EDL) were prepared as previously described[51]. Muscles were removed, and their tendons tied to wooden sticks to prevent contraction and placed in a relaxing solution composed of (mM): EGTA (5), $MgCl_2$ (2.5), $Na_2H_2ATP$ (2.5), imidazole (10), K-propionate (170), a protease inhibitor (1 Minitab per 10 mL, Roche), pH = 7. Over the next 18 h at 4 °C, 50% of the relaxing solution was gradually replaced with glycerol. Samples were then stored at −20 °C.

### Muscle mechanics
Length-dependence of $Ca^{2+}$ sensitivity was assessed by performing force-pCa curves at sarcomere lengths of 2.4 (short) and 2.8 (long) μm using standard protocols[52]. Steady-state active force was assessed at pCa's: 8.0, 6.33, 6.17, 6.0, 5.83, 5.67, 5.5, 5.0 (maximal $[Ca^{2+}]$). The solution contained (mM): EGTA (5), $MgCl_2$ (1.12), BES (20), $Na_2H_2ATP$ (5), Na-methyl sulfonate (67), PiOH (5), creatine phosphate (15), creatine kinase (300 U/mL), pH = 7. Force was normalized to the maximum force at pCa 5.0. A four-parameter Hill equation was fit to the normalized force-pCa data[53] to calculate $pCa_{50}$ (pCa at 50% maximum force; a measure of $Ca^{2+}$ sensitivity) and the Hill coefficient (Igor Pro; WaveMetrics, Oregon, USA). At pCa 5, a quick stretch of 0.25% muscle length was applied, and the force was fit to an equation of three exponentials: $A \exp(-k_{release}\ t) - B\exp(-k_{redevelopment}\ t) + C\exp(-k_{slow}\ t)$.

### Small angle X-ray diffraction and fiber mechanics apparatus
Samples were shipped to the BioCAT facility on ice for all experimental tests and stored at −20 °C until used. On the day of experiments, EDL muscles were removed from the storage solution and vigorously washed in relaxing solution (composition (in mM): potassium propionate (45.3), N,N-Bis(2-hydroxyethyl)-2-aminoethanesulfonic acid BES (40); EGTA (10), $MgCl_2$

(6.3), Na-ATP (6.1), DTT (10), protease inhibitors (complete), pH 7.0)). Whole fascicles were excised from EDL muscle and silk suture knots (sizing 6-0 or 4-0) were tied at the distal and proximal ends at the muscle-tendon junction as close to the fascicle as possible. Samples were then immediately transferred to the experimental chamber.

X-ray diffraction patterns were collected using the small-angle instrument on the BioCAT beamline 18ID at the Advanced Photon Source, Argonne National Laboratory[54]. The X-ray beam (0.103 nm wavelength) was focused to ~0.06 × 0.15 mm at the detector plane. X-ray exposures were set at 1 s with an incident flux of ~$3 \times 10^{12}$ photons per second. The sample-to-detector distance was set between 3.0 and 3.5 m, and the X-ray fiber diffraction patterns were collected with a CCD-based X-ray detector (Mar 165, Rayonix Inc., Evanston, IL, USA). An inline camera built into the system allowed for initial alignment with the X-ray beam and continuous sample visualization during the experiment. Prepared fiber bundles were attached longitudinally to a force transducer (402 A, Aurora Scientific, Aurora, Canada) and motor (322 C, Aurora Scientific, Aurora, Canada), and placed into a bath of relaxing solution at 27 °C. Force and length data were collected at 1000 Hz using a 600 A: Real-Time Muscle Data Acquisition and Analysis System (Aurora Scientific, Aurora, Canada). Sarcomere length (SL) was measured via laser diffraction using a 4-mW Helium–Neon laser. The force baseline was set at slack length = 0 mN. After this initial setup, fiber length changes were accomplished through computer control of the motor, which we confirmed appropriate SL length change on a subset of samples. The mechanical rig was supported on a custom-designed motorized platform that allowed placement of muscle into the X-ray flight path and small movements to target X-ray exposure during experiments. Using the inline camera of the X-ray apparatus, the platform was moved to target the beam at different locations along the length of the sample. To limit X-ray exposure of any one part of the preparation, no part of the sample was exposed more than once.

### Experimental protocols and analysis
The experimental approach captured X-ray images in samples at two SLs across the in vivo physiological operating range[55]. Samples were stretched from 2.4 μm SL to 2.8 μm SL at 0.1 μm SL s$^{-1}$ with a 90 s hold phase to allow for stress relaxation. X-ray images were collected at the end of each hold phase.

### Analysis of X-ray diffraction patterns
X-ray images were analyzed using the MuscleX open-source data reduction package[56]. The "Quadrant Folding" routine was used to improve the signal-to-noise by adding together the four equatorial-meridional quadrants, which each provide the same information (Friedel's Law). The "Equator" routine of MuscleX was used to calculate the $I_{1,1}/I_{1,0}$ intensity ratio, $d_{1,0}$ lattice spacing, and $\sigma_D$. Meridional (M3, T3, M6) and off-meridional reflections (A6) were analyzed using the MuscleX "Projection Traces" routine. Spacing measurements of the meridional reflections were made in the reciprocal radial range $\sim 0 \leq R \leq 0.032$ nm$^{-1}$ for M3, M6, and T3 reflections, and $\sim 0.013 \leq R \leq 0.053$ nm$^{-1}$ for the A6 reflection, where R denotes the radial coordinate in reciprocal space[57]. Every image provides intensities of different quality, which leads to various levels of Gaussian fit errors for each intensity modeled, which increases the variation in spacings in the dataset. To limit these effects, fit errors >10% were discarded. Positions of X-ray reflections on the diffraction patterns in pixels were converted to sample periodicities in nm using the 100-diffraction ring of silver behenate at $d_{001} = 5.8380$ nm. Intensity was normalized by the radially symmetric background measured by the "Quadrant Folding" routine.

### Statistics and reproducibility
Statistical analysis was conducted using JMP Pro (V16, SAS Institute, USA). The significance level was always set at α = 0.05. We used a repeated-measures analysis of variance (ANOVA) design with fixed effects SL, genotype, SL × genotype interaction term, and a nested random (repeated-measures) effect of the individual across SL levels. Data was best Box–Cox transformed to meet assumptions of normality and homoscedasticity when necessary, which were assessed by residual analysis, Shapiro–Wilk's test for normality, and Levene's test for unequal variance. Significant main effects were subject to Tukey's highly significant difference (HSD) multiple comparison procedure to assess differences between factor levels. This data is presented in figure panels via so-called connecting letters, where factor levels sharing a common letter are not significantly different from each other. All data presented as mean ± s.e.m. Sample sizes and further statistical details are presented in Tables 1 and 2.

## Data availability
Datasets used to generate the figures and tables are included as Supplementary Data 1. Any other data or materials are available from the corresponding authors on reasonable request.

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

## Acknowledgements

We thank the BioCAT beamline support staff at the APS for their steadfast commitment to our project and Anna Good for critical text and artistic editing. Figure cartoons in panels 1a and 2i were created using Biorender. This research used resources from the Advanced Photon Source, a U.S. Department of Energy (DOE) Office of Science User Facility operated for the

DOE Office of Science by Argonne National Laboratory under Contract No. DE - AC02 - 06CH11357, and further NIH support. The content is solely the responsibility of the authors and does not necessarily reflect the official views of the National Institute of General Medical Sciences or the National Institutes of Health. Funding was provided by the German Research Foundation grant 454867250 (A.L.H.), German Research Foundation grant SFB1002, A08 (W.A.L.), IZKF Münster Li1/029/20 (W.A.L.), National Institutes of Health P41 GM103622 (T.I.), P30 GM138395 (T.I.), R01 AR079435 (S.S.), R01 AR079477 (S.S.), R01 HL130356 (S.S.), R01 HL105826 (S.S.), R01 AR078001 (S.S.), R01 HL143490 (S.S.), and R01 HL150953 (B.M.P.), R01 HL171657 (W.M.), The American Heart Association 19UFEL34380251 (S.S.), 19TPA34830084 (S.S.) and 945748 (S.S.), The PLN Foundation (S.S.), and The American Heart Association Career Development Award (23CDA1046498) to T.S.

## Author contributions

Conceptualization: B.M.P., A.L.H. Methodology: A.L.H., B.M.P., B.A.M., T.S., S.S. Investigation: B.M.P., A.L.H., M.K., S.H., W.M. Visualization: A.L.H., B.M.P., W.M., W.L., T.C.I. Funding acquisition: B.M.P., A.L.H., T.C.I., T.S., S.S., W.L. Project administration: A.L.H., B.M.P. Supervision: A.L.H., B.M.P. Writing—original draft: A.L.H. Writing—review & editing: All authors.

## Funding

## Competing interests

The authors declare the following competing interests: W.M. and T.I. provides consulting and collaborative research studies to Edgewise Therapeutics Inc. A.L.H. and M.K. are owners of Accelerated Muscle Biotechnologies Consultants LLC, and S.S. provides consulting and collaborative research studies to the Leducq Foundation (CURE-PLAN), Red Saree Inc., Greater Cincinnati Tamil Sangam, Novo Nordisk, Pfizer, AavantiBio, Affinia Therapeutics Inc., Cardiocare Genetics—Cosmogene Skincare Pvt Ltd., AstraZeneca, MyoKardia, Merck and Amgen, but such work is unrelated to the content of this article. The remaining authors declare no competing interests.
