## [Peer Review File · Communications Biology]

Reviewers' comments:

Reviewer #1 (Remarks to the Author):

Dear Editors,

In "Fast myosin binding protein C knockout in skeletal muscle alters length-dependent activation and myofilament structure", Hessel and co-workers, address the role of the fast isoform of Myosin Binding Protein C (fMyBP-C) in regulating myosin head states within the sarcomere and its influence on muscle contractility.

This study exploits the homozygous fMyBP-C knock-out mouse (C2-/-) which was originally presented and characterized by Song et al. 2021 (10.1073/pnas.2003596118) to presented a follow-up study to another recent pre-print (Hessel et al. 2023 BioRxiv 10.1101/2023.09.10.556972). In that first pre-print, the authors reported observing that upon cleavage of the fMyBP-CC1C7 domains there was a sarcomere length (SL)-independent bias of the heads towards the ON state. At the same time, the SL-dependent transition towards the ON state remained intact. Based on these observations, the authors hypothesize that fMyBP-CC1C7 may regulate the SL-independent control of the ON/OFF state, while fMyBP-CC8C10 may regulate the SL-dependent control of the heads state.

In this study under review in Comm. Bio, the authors tested this hypothesis by comparing the global fMyBP-C knockout mouse (C2-/-) to a non-transgenic (NTG) control.

The authors meticulously examined the mechanical functionality and the myofilament structure of their models while providing scrupulous description of their experimental methods.

The conclusions can be summarized as follow:

1. The C2-/- fiber present deficits in force production and calcium sensitivity.
2. The C2-/- fiber show altered SL-independent and SL-dependend regulation of the head - favoring the ON state.
3. Additionally, at short SL, the C2-/- fiber present axially extended thin filaments.

The authors confirm and reinforce the reduced force production and calcium sensitivity reported in Song et al. 2021 while also providing validation to the observations of the SL-independent role of MyBP-C (Hessel et al. 2023 BioRxiv 10.1101/2023.09.10.556972). Additionally, they provide novel insights into the SL-dependent role of MyBP-C in the regulation of the myosin heads. The third conclusion regards thin filament extension and contributes with novel and thought-provoking data on this poorly understood subject.

These conclusions on the C2-/- KO fibers are solid, in agreement with previous studies, and with added value. The interpretations of the observations and the inferred role of fMyBP-C could benefit from further clarification.

Hessel and co-workers studied the extensor digitorum longus (EDL), which contains both slow and fast MyBP-C isoforms and presents a 43% fMyBP-C expression. As a result, the C2-/- KO mouse has approximately 24% fewer MyBP-C, while also showing a significant increase (30%, Song et al. 2021) in sMyBP-C levels. The authors assume their observations to be a direct effect of the reduced abundance of

MyBP-C. However, given the known structural differences in the N-terminus domains of fast vs slow MyBP-C isoform, how can the authors exclude that all their readings are not the consequence of an increased relative abundance of sMyBP-CC1C7 in the C2-/- compared to the NTG?

Related to this question, the authors suggest that the increased radial distance between myofilament (observed in the C2-/-) can be associated to the exclusive presence of sMyBP-C rather than to be dependent on the overall decreased levels of MyBP-C. Couldn't this same line of reasoning be applied to the other observations? Is there any additional information available on this subject to support the statement?

The conclusion that SL-extension did not elongate the thick filament or reorient the myosin heads can be associated, as the authors point out, to the relative weak difference in signal. Could the authors analyze the behaviors of the C2-/- model in presence of compounds (e.g. mavacamten) that could stabilise the OFF state? These observations could potentially help to better pinpoint and dissect the role of fMyBP-C KO in SL-dependent and SL-independent mechanisms and significantly strengthen the proposed narrative for the role of fMyBP-C.

Additional comments

Line 5: Consider clarifying. e.g. "Myosin-binding protein C (MyBP-C) is bound to the myosin thick filament and is predicted to stabilize myosin heads in a docked position against the thick filament, the OFF state, which limits crossbridge formation."

Line 10: The abbreviation "SL" has not been introduced yet.

Line 41-43: please rephrase for readability.

"(...)interactions with up to 108 myosin heads per thick filament^{22,23}, with MyBP-C dysfunction associated with debilitating human myopathies"

Line 59: the sentence "with a modest increase in sMyBP-C" might be misleading: the referenced study reported a 30% increase in sMyBP-C level. See line 136, too.

Line 112: please invert to "(SM3 / IM3)" for consistency and readability.

Line 136: the authors initially report a 24% total decrease in MyBP-C level in their model. However, this line suggests having removed "about 50% of the MyBP-C molecules from the thick filament". Could you please elaborate?

Reviewer #2 (Remarks to the Author):

Song et al. 2021 described a novel knockout mouse model of fast skeletal myosin-binding protein C

fMyBP-C (1). In the current study, the authors investigated length-dependent activation in extensor digitorum longus (EDL) muscles from that mouse model vs. age-matched non-transgenic (NTG) controls. They conducted carefully designed mechanical and X-ray diffraction experiments, which were analyzed well using appropriate statistical analysis. They reported some interesting sarcomere length-dependent and sarcomere length-independent effects, which have not been characterized before in that mouse model. Some of the same authors also have another study in preprint that investigates the effect of removal of the N-terminal first seven domains of fMyBP-C on some of the same mechanical and X-ray parameters at different sarcomere lengths in fast-twitch dominant psoas muscle (2). There are not as many studies on skeletal MyBP-C isoforms, hence their findings will be of interest to a specialized audience.

There are some interesting effects, particularly the effect on the myosin head conformations. However, the authors sometimes overinterpret their results. The actual mouse model is relatively mild with no signs of excess morbidity and mortality, and some of that may be due to compensatory changes during development. In EDL muscle, there is a mixture of slow and fast skeletal MyBP-C and in the knockout mice, the lack of the fast isoform is compensated to some level, albeit not fully, by increased expression of the slow isoform (+33%). In addition, the total levels of the regulatory light chain of myosin were significantly decreased (-26%) (1). There is not enough explanation how sMyBP-C and reduced RLC levels can contribute to the observed effects.

The authors should revise their manuscript and consider the following points:

Major

1. Where are the data points in Figure 1 B and C? I would like to see the points and SD, if not in a main figure, than at least in a supplemental figure. Where is the supplement mentioned in line 277?
2. How may MLC2 protein downregulation in C2-/- EDL muscles contribute to the measured effects?
3. In EDL, there is a compensatory increase in sMyBP-C levels compared to wt (33% higher expression of sMyBP-C in EDL), so the difference with wt is not only in lack of fMyBP-C but also in the levels of sMyBP-C. The question arises to what extent the effects of fMyBP-C removal are rescued or modulated by increased sMyBP-C expression? In that sense the removal of the N-terminal C1C7 domains is a cleaner experiment and can help to interpret the results in this study. The conclusions in this study need to be more carefully phrased to reflect that there is an uncertainty about the functional contributions of compensatory changes. This can be addressed in future studies using sMyBP-C knock-out and double sMyBP-C and fMyBP-C knock-out.
4. In Table 2, the genotype, SL and interaction effects for d1,0 have a $p < 0.05$. However, after Tukey's highly significant difference (HSD) multiple comparison the only significant effect is due to SL. In the text line 89, it says that there is no genotype effect for inter-filament spacing but in lines 94-96, it says that sMyBP-C supports a greater radial distance between myofilaments compared to fMyBP-C. There must be an error, otherwise the last statement cannot be true.

Minor

1. Lines 12-14 postulate is too strongly worded. It should be hypothesize. This hypothesis can be tested by inhibiting myosin and checking for thin filament axial extension in that model.
2. Lines 35-37. It needs to be toned down and specified and even though this paper is about skeletal muscle, a reader may assume it as a general statement. There is no consensus that this is the one and only mechanism underpinning length dependent activation or that it works in the same way in different mammalian muscle types but there is strong evidence for mechanosensing during resting stretch in skeletal muscle (3, 4).

3. It would help to improve understanding, if the mouse knock-out model is introduced earlier, somewhere after line 48.
4. Line 52-53. It will be clearer for the reader: they hypothesize that the C1-C7 domains regulate the SL-independent myosin head ON/OFF transitions, while C8-C10 domains regulate the SL-dependent transitions.
5. Line 62-63. It would help understanding to specify the effect on the SL-dependent structural changes or using a different word than altered (diminished?).
6. Line 79-82: something is missing. "We found no genotype effects for the rate of force release (KRelease), force redevelopment (KRedevelopment), or the slow phase (Kslow) from the force redevelopment curve (see methods) were all decreased at the longer vs. shorter SL (Fig. 1I-K;Table 1), as expected." Do the authors mean that muscles from the knockout showed decreased rates at long vs. short SL same as in the wild type muscles?
7. Line 99-102. Sarcomere length-dependent mechanisms are not fully elucidated. In resting skeletal muscle, stretch increases thick filament stress and triggers the release of the myosin motors from their folded conformation by a mechano-sensing mechanism (5). In cardiac muscle, the picture is less clear. Some x-ray diffraction studies have shown that stretch does not affect the OFF state of the thick filament in intact cardiac muscle during diastole (6), while others have found structural changes in thick filaments (7).
8. Line 125 and line 146, do the authors mean $I_{1,1}/I_{1,0}$?
9. Line 159, suggestion: by an unknown mechanism.
10. Line 174, do the authors mean weak-binding crossbridges?
11. Line 180-181. Suggested edit: It is not known how this pool of weak-binding myosin crossbridges and changes in thin filament length modulate muscle contraction.

1. T. Song et al., Fast skeletal myosin-binding protein-C regulates fast skeletal muscle contraction. *Proc Natl Acad Sci U S A* 118 (2021).
2. A. L. Hessel et al., Myosin-binding protein C forms C-links and stabilizes OFF states of myosin. *bioRxiv* 10.1101/2023.09.10.556972 (2023).
3. T. C. Irving, R. Craig, Getting into the thick (and thin) of it. *J Gen Physiol* 151, 610-613 (2019).
4. L. Fusi, E. Brunello, Z. Yan, M. Irving, Thick filament mechano-sensing is a calcium-independent regulatory mechanism in skeletal muscle. *Nat Commun* 7, 13281 (2016).
5. M. Irving, Regulation of Contraction by the Thick Filaments in Skeletal Muscle. *Biophys J* 113, 2579-2594 (2017).
6. M. Caremani et al., Inotropic interventions do not change the resting state of myosin motors during cardiac diastole. *J Gen Physiol* 151, 53-65 (2019).
7. Y. Ait-Mou et al., Titin strain contributes to the Frank-Starling law of the heart by structural rearrangements of both thin- and thick-filament proteins. *Proc Natl Acad Sci U S A* 113, 2306-2311 (2016).

Reviewer #3 (Remarks to the Author):

This is a nice study, building on the authors' expertise on this subject.

Summary

In this study, Dr Hessel and team have used the fast skeletal muscle from a mouse with fast-MyBP-C-ko, labelled C2+/-, from the lab of Sadayappan. Fast skeletal fibres have both fast and slow MyBP-C, hence the C2-/- muscles have only slow MyBP-C (~20% in EDL muscle).

Here Hessel and team performed mechanical experiments and carried out X-ray analysis of demembrated EDL NTG and C2-/- fibres at two different sarcomere lengths, 2.4 and 2.8 μm . This is a good choice of SL as it is within the range where linear effects were observed in previous studies (eg Hessel et al, PNAS, 2022).

They report reduced Ca sensitivity and reduced active tension for the ko fibres. X-ray analysis shows higher I11/10 values for the ko fibres, in agreement with previous studies. They examined periodicities of thick and thin filaments, observing increased values for both filaments NTG fibres but little change in the ko.

Comments

I found this a hard paper to review because of the lack of care in the writing in some places.

To talk about movement of mass from myosin to actin, we refer to the ratio I11/10 as in the axis label in Fig 2e. So why not use this consistently rather than refer to I10/11 as in the text on line 125 and line 145. I10/11 gives the inverse ratio which I presume was not the intention of the authors. On line 151 the ratio is stated correctly.

What I found troublesome is the major error for the captions in Figure 1 and 2, that C2-/- is represented by black/squares and NTG by red/triangles. This is wrong and the reverse is correct. If the authors knew about it after submission, they should have immediately informed the editorial office so that the reviewers could be informed. In the key for Fig 1b and c, the correct colour convention is used, black for NTG and red for ko. I wasted a lot of time grappling with this.

Discussion about X-ray evidence for thick and thin filament length change.

The authors have examined reflections originating from the thick and thin filaments and they report an increase in nm in the spacings of these reflections. On the other hand, it was reported long ago that upon activation, the thick filament undergoes an increase of ~1%. I think it is misleading to talk about change in filament length per se as it implies a change in the actual length as in skeletal versus cardiac muscles (see Granzier et al, Am J Physiol, 1991). A small change like <1% can be brought about by change in the molecular packing. I think the authors should also report the changes as a percentage change of the total length of the filaments.

Minor comments

In both figures, the black squares and red triangles are hard to see because of their fine hairline outlines. I suggest the authors use coloured filled-in circles for the points in the graph eg like Fig 4c in Song et al, Fast skeletal MyBP-C regulates fast skeletal muscle, PNAS, 2021. Please note that saturated colours in tiny graphs like Fig 4c of Song are much easier to see.

Line 67 Re fig 1 b and c

In figure legend, clarify which is the dashed line, by drawing a dashed line with two dashes

Line 75 Fig 1f, SL-independent reduction in Ca sensitivity

Fig 1h Put key in the figure for the colours: orange=2.4um

Line 155-157 This is a very obscure sentence, please clarify.

Line 438 In the I-band, titin extends from the Z-disk to the tops of the ...

Surely it should be "tips" of the thick filament

Below we provide the original decision letter (reviewer comments portion only) and our replies in a red font. Changes to the text that are reproduced here are indicated by a blue font. Within the manuscript, changes are shown in red. In the rebuttal letter only, we show (name, year) citation style, so readers can quickly identify the citation.

Reviewer #1 (Remarks to the Author):

Dear Editors,

In "Fast myosin binding protein C knockout in skeletal muscle alters length-dependent activation and myofilament structure", Hessel and co-workers, address the role of the fast isoform of Myosin Binding Protein C (fMyBP-C) in regulating myosin head states within the sarcomere and its influence on muscle contractility. This study exploits the homozygous fMyBP-C knock-out mouse (C2^{-/-}) which was originally presented and characterized by Song et al. 2021 (10.1073/pnas.2003596118) to presented a follow-up study to another recent pre-print (Hessel et al. 2023 BioRxiv 10.1101/2023.09.10.556972). In that first pre-print, the authors reported observing that upon cleavage of the fMyBP-CC1C7 domains there was a sarcomere length (SL)-independent bias of the heads towards the ON state. At the same time, the SL-dependent transition towards the ON state remained intact. Based on these observations, the authors hypothesize that fMyBP-CC1C7 may regulate the SL-independent control of the ON/OFF state, while fMyBP-CC8C10 may regulate the SL-dependent control of the heads state.

In this study under review in Comm. Bio, the authors tested this hypothesis by comparing the global fMyBP-C knockout mouse (C2^{-/-}) to a non-transgenic (NTG) control. The authors meticulously examined the mechanical functionality and the myofilament structure of their models while providing scrupulous description of their experimental methods.

The conclusions can be summarized as follow:

1. The C2^{-/-} fiber present deficits in force production and calcium sensitivity.
2. The C2^{-/-} fiber show altered SL-independent and SL-depended regulation of the head - favoring the ON state.
3. Additionally, at short SL, the C2^{-/-} fiber present axially extended thin filaments.

The authors confirm and reinforce the reduced force production and calcium sensitivity reported in Song et al. 2021 while also providing validation to the observations of the SL-independent role of MyBP-C (Hessel et al. 2023 BioRxiv 10.1101/2023.09.10.556972). Additionally, they provide novel insights into the SL-dependent role of MyBP-C in the regulation of the myosin heads. The third conclusion regards thin filament extension and contributes with novel and thought-provoking data on this poorly understood subject. These conclusions on the C2^{-/-} KO fibers are solid, in agreement with previous studies, and with added value. The interpretations of the observations and the inferred role of fMyBP-C could benefit from further clarification.

Hessel and co-workers studied the extensor digitorum longus (EDL), which contains both slow and fast MyBP-C isoforms and presents a 43% fMyBP-C expression. As a result, the C2^{-/-} KO mouse has approximately 24% fewer MyBP-C, while also showing a significant increase (30%, Song et al. 2021) in sMyBP-C levels. The authors assume their observations to be a direct effect of the reduced abundance of MyBP-C. However, given the known structural differences in the N-terminus domains of fast vs slow MyBP-C isoform, how can the authors exclude that all their readings are not the consequence of an increased relative abundance of sMyBP-CC1C7 in the C2^{-/-} compared to the NTG?

The reviewer touches on a point that the authors debated rigorously but did not get much treatment in the original manuscript. We now discuss these possibilities more thoroughly. Generally, we note that the different isoforms of MyBP-C can differentially affect the activation of the thin filament even under non-activating calcium conditions. As noted by Li et al. (Li et al., 2019) there is at least one alternative splicing of slow MyBP-C that activates the thin filament of EDL much more so than the fast isoform or other alternative splicings. Relevant to our work, these previously reported findings suggest that, although a loss of total MyBP-C in the C2^{-/-} would appear the most likely basis of differential results, an enhanced activation of the thin filament by the upregulation of slow MyBP-C cannot be completely ruled out. Still, if there were increased interaction of MyBP-C with the thin filament in the C2^{-/-} and possibly some activation of the thin filament during the X-ray measures, we would expect as a consequence enhanced thick filament elongation and enhanced compression of the lattice compared to what we found for the C2^{-/-}. Therefore, the effects of increased slow MyBP-C would not likely underlie the structural results reported here, but could partially mask the effects of the loss of MyBP-C. Nevertheless, these are all ideas worth evaluating in the future.

We now update our discussion section to include these points.

From line 99:

“The C2^{-/-} fibers have a relatively complex MyBP-C landscape, with no fMyBP-C and upregulated sMyBP-C that nonetheless leads to a ~50% reduction of total MyBP-C content in the mouse EDL. In addition, sMyBP-C and fMyBP-C have well-characterized structural differences in their N-termini that lead to different functionalities that are still not fully described (Li et al., 2019; Song et al., 2023). Follow-up studies with sMyBP-C KO and f+sMyBP-C KOs would help parse out further details. What can be said currently is that the altered MyBP-C composition impacts lattice order across SLs, suggesting MyBP-C’s ability to interact via thin filament interactions, as recently demonstrated (Harris, 2021; Huang et al., 2023; Tamborrini et al., 2023).”

From line 163:

“What we cannot yet infer from this study is if the removal of fMyBP-C is the sole driver of these changes, or if we are observing the effects of a general decrease in the total MyBP-C. Furthermore, sMyBP-C, with its diverse functionality (Ackermann and Kontrogianni-Konstantopoulos, 2011; Ackermann and Kontrogianni-Konstantopoulos, 2013), may rescue the phenotype if it was upregulated to account for all missing fMyBP-C molecules. sMyBP-C or s+fMyBP-C knockouts would help elucidate this lingering question.”

More generally at the end of the introduction, we indicate other potential (and not necessarily mutually exclusive) mechanisms causing the effects of the C2^{-/-}.

From line 66:

“The underlying mechanisms for this outcome are potentially related to several processes that are not necessarily mutually exclusive, including loss of fMyBP-C, decrease in total MyBP-C content, and/or increased expression of sMyBP-C (Song et al., 2021).”

Related to this question, the authors suggest that the increased radial distance between myofilament (observed in the C2^{-/-}) can be associated to the exclusive presence of sMyBP-C rather than to be dependent on the overall decreased levels of MyBP-C. Couldn’t this same line of reasoning be applied to the other observations? Is there any additional information available on this subject to support the statement?

The reviewer points out that changes to lattice spacing and other parameters could just be caused by a decrease in the proportion of MyBP-C content (slow + fast) and may not provide additional evidence that slow and fast MyBP-C isoforms function differently. In this respect, we agree. As we suggest in the Discussion, we expect the loss of MyBP-C content to be the underlying explanation for our findings. Furthermore, we are unclear what exact role the slow isoforms play in this regard. If we wanted to parse out a specific response by fast or slow isoforms, it would be worthwhile to conduct similar experiments of psoas muscle with just a slow MyBP-C KO, and then also one with a slow+fast MyBP-C KO. As discussed in the previous comment above, we now present these possibilities.

From line 99:

“The C2^{-/-} fibers have a relatively complex MyBP-C landscape, with no fMyBP-C and upregulated sMyBP-C that nonetheless leads to a ~50% reduction of total MyBP-C content in the mouse EDL. In addition, sMyBP-C and fMyBP-C have well-characterized structural differences in their N-termini that lead to different functionalities that are still not fully described (Li et al., 2019; Song et al., 2023). Follow-up studies with sMyBP-C KO and f+sMyBP-C KOs would help parse out further details. What can be said currently is that the altered MyBP-C composition impacts lattice order across SLs, suggesting MyBP-C’s ability to interact via thin filament interactions, as recently demonstrated (Harris, 2021; Huang et al., 2023; Tamborrini et al., 2023).”

The conclusion that SL-extension did not elongate the thick filament or reorient the myosin heads can be associated, as the authors point out, to the relative weak difference in signal. Could the authors analyze the behaviors of the C2^{-/-} model in presence of compounds (e.g. mavacamten) that could stabilise the OFF state? These observations could potentially help to better pinpoint and dissect the role of fMyBP-C KO in SL-dependent and SL-independent mechanisms and significantly strengthen the proposed narrative for the role of fMyBP-C.

We agree that follow-up studies like this will be informative. The use of mavacamten, as well as myosin activators such as dATP are something that we are interested in for future studies. X-ray studies like this come at a considerable time and cost burden, of which we are currently writing grants for. We hope these details will be elucidated over the next couple of years.

From line 177:

“The relationship of the M3, M6, and I₁₁/I₁₀ in C2^{-/-} could be further studied by forcibly shifting myosin heads into the OFF or ON states using (for example) the myosin deactivator mavacamten (Ma et al., 2023b) or myosin activator dATP (Ma et al., 2020; Ma et al., 2023a). Taken together, we find that compared to NTG, the measured alterations of myosin head regulation in the C2^{-/-} can be explained by a partial destabilization of the OFF state.”

Additional comments

Line 5: Consider clarifying. e.g. “Myosin-binding protein C (MyBP-C) is bound to the myosin thick filament and is predicted to stabilize myosin heads in a docked position against the thick filament, the OFF state, which limits crossbridge formation.”

Done as suggested.

Line 10: The abbreviation “SL” has not been introduced yet.

Now spelled out on first use.

Line 41-43: please rephrase for readability.

“(…)interactions with up to 108 myosin heads per thick filament^{22,23}, with MyBP-C dysfunction associated with debilitating human myopathies”

Rephrased and now reads (From line 36):

“MyBP-C arises at ~43 nm intervals along the thick filament backbone, interacting with up to 108 myosin heads per thick filament (Dutta et al., 2023; Tamborrini et al., 2023). Of particular interest, MyBP-C dysfunction is associated with debilitating human myopathies linked to altered force production (Adhikari et al., 2019; Geist and Kontrogianni-Konstantopoulos, 2016; Gurnett et al., 2010; Sarkar et al., 2020).”

Line 59: the sentence “with a modest increase in sMyBP-C” might be misleading: the referenced study reported a 30% increase in sMyBP-C level. See line 136, too.

Understood. We now rephrase these sentences.

From line 56:

“The NTG EDL is predominately a fast-twitch muscle with a ~43% fMyBP-C expression (Li et al., 2019). C2^{-/-} express trace levels of fMyBP-C with a ~30% increase in sMyBP-C that leads to thick filaments with ~50% reduction in total MyBP-C molecules (Song et al., 2021).”

From line 151:

“MyBP-C appears to positions the subpopulation of myosin heads in the C-zone into a different orientation than those in the P- or D-zones (Reconditi, 2006). In C2^{-/-} fibers, there is a ~50% reduction in total MyBP-C molecules per thick. All the fast-MyBP-C molecules have been removed, accompanied by a 30% increase in slow MyBP-C that cannot account for all lost fast-isoform molecules.”

Line 112: please invert to “(SM3 / IM3)” for consistency and readability.

Corrected.

Line 136: the authors initially report a 24% total decrease in MyBP-C level in their model. However, this line suggests having removed “about 50% of the MyBP-C molecules from the thick filament”. Could you please elaborate?

This was a mistake. ~50% reduction in total MyBP-C content is correct.

The line now reads (from line 153):

“In C2^{-/-} fibers, there is a ~50% reduction in total MyBP-C molecules per thick. All the fast-MyBP-C molecules have been removed, accompanied by a 30% increase in slow MyBP-C that cannot account for all lost fast-isoform molecules.”

Reviewer #2 (Remarks to the Author):

Song et al. 2021 described a novel knockout mouse model of fast skeletal myosin-binding protein C fMyBP-C (1). In the current study, the authors investigated length-dependent activation in extensor digitorum longus (EDL) muscles from that mouse model vs. age-matched non-transgenic (NTG) controls. They conducted carefully designed mechanical and X-ray diffraction experiments, which were analyzed well using appropriate statistical analysis. The reported some interesting sarcomere length-dependent and

sarcomere length-independent effects, which have not been characterized before in that mouse model. Some of the same authors also have another study in preprint that investigates the effect of removal of the N-terminal first seven domains of fMyBP-C on some of the same mechanical and X-ray parameters at different sarcomere lengths in fast-twitch dominant psoas muscle (2). There are not as many studies on skeletal MyBP-C isoforms, hence their findings will be of interest to a specialized audience. There are some interesting effects, particularly the effect on the myosin head conformations. However, the authors sometimes overinterpret their results. The actual mouse model is relatively mild with no signs of excess morbidity and mortality, and some of that may be due to compensatory changes during development.

In EDL muscle, there is a mixture of slow and fast skeletal MyBP-C and in the knockout mice, the lack of the fast isoform is compensated to some level, albeit not fully, by increased expression of the slow isoform (+33%). In addition, the total levels of the regulatory light chain of myosin were significantly decreased (-26%) (1). There is not enough explanation how sMyBP-C and reduced RLC levels can contribute to the observed effects.

The request for more consideration on the role of sMyBP-C and RLC (specifically MYLC2) also came from the other reviewers. We now point out very clearly some of these other competing factors that may be affecting the sarcomeric structures, apart from just the effect caused purely by a fMyBP-C KO. We now note that the different isoforms of MyBP-C can differentially affect the activation of the thin filament under non-activating calcium conditions. As noted by Li et al. (Li et al., 2019) there is at least one alternative splicing of slow MyBP-C that activates the thin filament of EDL much more so than the fast isoform or other alternative splicings. Still, if there were increased interaction of MyBP-C with the thin filament in the C2^{-/-} and possibly some activation of the thin filament during the X-ray measures, we would expect enhanced thick filament elongation and enhanced compression of the lattice compared to what we found for the C2^{-/-}. Therefore, the effects of increased slow MyBP-C would not likely underlie the structural results reported here, but could partially mask the effects of the loss of MyBP-C.

With regards to the 35% reduction in Myl2 content in the C2^{-/-} as reported by Song et al., (2021), it should be noted that Myl2 is not the predominate myosin regulatory light chain (RLC) found in mouse EDL. The overwhelmingly predominate RLC isoform in mouse EDL is Myl1f, and the expression and protein content of Myl2 is comparatively negligible (Hettige et al., 2020; Sitbon et al., 2020). The loss of 35% of Myl2 would not be expected to have any structural or functional effect on the measures we report in the EDL.

A similar argument can be made for the reported upregulation of Myh3, the embryonic myosin heavy chain, also reported in the EDL of the C2^{-/-} mouse. The expression of Myh3 is very important for muscle development prior to adulthood and was upregulated by ~50% in the EDL of adult C2^{-/-} mice. However, considering it is negligibly expressed in adult mouse EDL compared to Myh4, myosin IIb (Hettige et al., 2020; Schiaffino et al., 2015; Song et al., 2021), the elevated expression of Mhc3 would still be a negligible quantity and not have any structural or functional effect on our measures.

We now bring this discussion to the reader.

From line 57:

“To explore the functional role of MyBP-C in skeletal muscle, we studied extensor digitorum longus (EDL) muscles from a fMyBP-C global knockout mouse (C2^{-/-}) vs. age-matched non-transgenic (NTG) controls (Song et al., 2021). The NTG EDL is predominately a fast-twitch muscle with a ~43% fMyBP-C expression (Li et al., 2019). C2^{-/-} express trace levels of fMyBP-C with a ~30% increase in sMyBP-C that leads to thick filaments with ~50% reduction in total MyBP-C molecules (Song et al., 2021). This model

also presents a reduction in one myosin regulatory light chain isoform (Mylc2) (Song et al., 2021), however, a different myosin light chain isoform (phosphorylatable, fast isoform; Mylpf) is overwhelmingly expressed in EDL fibers (Hettige et al., 2020; Sitbon et al., 2020) and is not altered in C2^{-/-} (Song et al., 2021), so a strong effect on sarcomere structure and function are unlikely.”

The authors should revise their manuscript and consider the following points:

Major

1. Where are the data points in Figure 1 B and C? I would like to see the points and SD, if not in a main figure, than at least in a supplemental figure.

We are in agreement that data points should be shown. In this case, panels B and C would not be easily legible with data points. Therefore, we added a supplemental figure with the individual datapoints (Supplemental Figure 1, shown below).

Where is the supplement mentioned in line 277?

This was in reference to the source dataset. It was not included in the original submission but is now available.

2. How may MLC2 protein downregulation in C2^{-/-} EDL muscles contribute to the measured effects?

As detailed above, with regards to the 35% reduction in Myl2 content in the C2^{-/-} as reported by Song et al., (2021), it should be noted that - unlike cardiac muscle or soleus - Myl2 is not the predominate myosin regulatory light chain found in mouse EDL. The overwhelmingly predominate RLC isoform in mouse EDL is Mylpf, and the expression and protein content of Myl2 is comparatively negligible (Hettige et al., 2020; Sitbon et al., 2020). The loss of 35% of Myl2 would not be expected to have any structural or functional effect on the measures we report.

From line 57:

“To explore the functional role of MyBP-C in skeletal muscle, we studied extensor digitorum longus (EDL) muscles from a fMyBP-C global knockout mouse ($C2^{-/-}$) vs. age-matched non-transgenic (NTG) controls (Song et al., 2021). The NTG EDL is predominately a fast-twitch muscle with a ~43% fMyBP-C expression (Li et al., 2019). $C2^{-/-}$ express trace levels of fMyBP-C with a ~30% increase in sMyBP-C that leads to thick filaments with ~50% reduction in total MyBP-C molecules (Song et al., 2021). This model also presents a reduction in one myosin regulatory light chain isoform (Mylc2) (Song et al., 2021), however, a different myosin light chain isoform (phosphorylatable, fast isoform; Mylpf) is overwhelmingly expressed in EDL fibers (Hettige et al., 2020; Sitbon et al., 2020) and is not altered in $C2^{-/-}$ (Song et al., 2021), so a strong effect on sarcomere structure and function are unlikely.”

3. In EDL, there is a compensatory increase in sMyBP-C levels compared to wt (33% higher expression of sMyBP-C in EDL), so the difference with wt is not only in lack of fMyBP-C but also in the levels of sMyBP-C. The question arises to what extent the effects of fMyBP-C removal are rescued or modulated by increased sMyBP-C expression? In that sense the removal of the N-terminal C1C7 domains is a cleaner experiment and can help to interpret the results in this study. The conclusions in this study need to be more carefully phrased to reflect that there is an uncertainty about the functional contributions of compensatory changes. This can be addressed in future studies using sMyBP-C knock-out and double sMyBP-C and fMyBP-C knock-out.

The reviewer brings up a good point, and one that is also discussed by reviewer 1. We now provide more discussion on the various potential underlying mechanisms that lead to our findings. We also consider future directions that could provide more insight.

From line 54 (introduction):

To explore the functional role of MyBP-C in skeletal muscle, we studied extensor digitorum longus (EDL) muscles from a fMyBP-C global knockout mouse ($C2^{-/-}$) vs. age-matched non-transgenic (NTG) controls (Song et al., 2021). The NTG EDL is predominately a fast-twitch muscle with a ~43% fMyBP-C expression (Li et al., 2019). $C2^{-/-}$ express trace levels of fMyBP-C with a ~30% increase in sMyBP-C that leads to thick filaments with ~50% reduction in total MyBP-C molecules (Song et al., 2021). This model also presents a reduction in one myosin regulatory light chain isoform (Mylc2) (Song et al., 2021), however, a different myosin light chain isoform (phosphorylatable, fast isoform; Mylpf) is overwhelmingly expressed in EDL fibers (Hettige et al., 2020; Sitbon et al., 2020) and is not altered in $C2^{-/-}$ (Song et al., 2021), so a strong effect on sarcomere structure and function are unlikely.”

From line 99 (discussion):

“The $C2^{-/-}$ fibers have a relatively complex MyBP-C landscape, with no fMyBP-C and upregulated sMyBP-C that nonetheless leads to a ~50% reduction of total MyBP-C content in the mouse EDL. In addition, sMyBP-C and fMyBP-C have well-characterized structural differences in their N-termini that lead to different functionalities that are still not fully described (Li et al., 2019; Song et al., 2023). Follow-up studies with sMyBP-C KO and f+sMyBP-C KOs would help parse out further details. What can be said currently is that the altered MyBP-C composition impacts lattice order across SLs, suggesting MyBP-C’s ability to interact via thin filament interactions, as recently demonstrated (Harris, 2021; Huang et al., 2023; Tamborrini et al., 2023).

From line 160 (discussion):

“What we cannot yet infer from this study is if the removal of fMyBP-C is the sole driver of these changes, or if we are observing the effects of a general decrease in the total MyBP-C. Furthermore, sMyBP-C, with its diverse functionality (Ackermann and Kontrogianni-Konstantopoulos, 2011; Ackermann and Kontrogianni-Konstantopoulos, 2013), may rescue the phenotype if it was upregulated to account for all missing fMyBP-C molecules. sMyBP-C or s+fMyBP-C knockouts would help elucidate this lingering question.”

4. In Table 2, the genotype, SL and interaction effects for d1,0 have a $p < 0.05$. However, after Tukey’s highly significant difference (HSD) multiple comparison the only significant effect is due to SL. In the text line 89, it says that there is no genotype effect for inter-filament spacing but in lines 94-96, it says that sMyBP-C supports a greater radial distance between myofilaments compared to fMyBP-C. There must be an error, otherwise the last statement cannot be true.

This was a data entry error and fixed.

Minor

1. Lines 12-14 postulate is too strongly worded. It should be hypothesize. This hypothesis can be tested by inhibiting myosin and checking for thin filament axial extension in that model.

Changed as suggested.

From line 198:

“In this study, $C2^{-/-}$ sarcomeres have more ON myosin heads as well as longer thin filaments. In passive muscles, ON myosins produce a small number of force-producing crossbridges that generate a small amount of force on the thin filaments (Donaldson et al., 2012; Eakins et al., 2016; Jarvis et al., 2021; Regnier et al., 1995). These bound crossbridges, estimated at ~2% of myosin heads, increase with increasing proportion of ON myosin heads (Prodanovic et al., 2023). In $C2^{-/-}$ fibers, more myosin heads are ON (see above), and so we hypothesize that more force-producing crossbridges are engaged as well, contributing to thin filament extension. This hypothesis can be tested by using mavacamten or dATP to force nearly all myosin heads into the OFF or ON state, respectively (Ma et al., 2023a; Ma et al., 2023b), and tracking changes to thin filament length. It is not known how this increased pool of force-producing myosin crossbridges and changes in thin filament length modulate muscle contraction.”

2. Lines 35-37. It needs to be toned down and specified and even though this paper is about skeletal muscle, a reader may assume it as a general statement. There is no consensus that this is the one and only mechanism underpinning length dependent activation or that it works in the same way in different mammalian muscle types but there is strong evidence for mechanosensing during resting stretch in skeletal muscle (3, 4).

We toned it down and made it more specific, as suggested.

From line 28:

“In skeletal muscle, there is evidence for a sarcomere length (SL)-dependent transition of myosin heads, where increasing SL shifts myosin heads toward the ON state and is thought to be an important mechanical underpinning of length-dependence of calcium sensitivity (length-dependent activation) (de Tombe et al., 2010; Fusi et al., 2016; Irving and Craig, 2019; Linari et al., 2015; Ma et al., 2018).”

3. It would help to improve understanding, if the mouse knock-out model is introduced earlier, somewhere after line 48.

Done as suggested, and we modified the rest of the section to make sense with the new order of thought (also comment below).

4. Line 52-53. It will be clearer for the reader: they hypothesize that the C1-C7 domains regulate the SL-independent myosin head ON/OFF transitions, while C8-C10 domains regulate the SL-dependent transitions.

Done as suggested.

5. Line 62-63. It would help understanding to specify the effect on the SL-dependent structural changes or using a different word than altered (diminished?).

Done as suggested. It now reads (from line 66):

“C2^{-/-} fibers showed an SL-independent shift toward the ON state while SL-dependent structural changes in thick and thin filament periodicities were either diminished or not detectable.”

6. Line 79-82: something is missing. “We found no genotype effects for the rate of force release (KRelease), force redevelopment (KRedevelopment), or the slow phase (Kslow) from the force redevelopment curve (see methods) were all decreased at the longer vs. shorter SL (Fig. 1I-K; Table 1), as expected.” Do the authors mean that muscles from the knockout showed decreased rates at long vs. short SL same as in the wild type muscles?

We have fixed this typo. The reviewer is correct. From line 111:

“It is important to note that these ideas are actively evolving. While there is consensus that the spacing of the M6 reflection (S_{M6}) reflects the periodicity of structures within the thick filament backbone, where these small changes in thick filament backbone periodicity (<1%) are due to actual extension of molecules themselves, or changes in the molecular packing, or both, has been a subject of considerable debate and is not yet clear (Dutta et al., 2023; Huxley et al., 2003; Zoghbi et al., 2008). Similarly, while the OFF to ON transitions seem to be associated with passive stretch in skeletal muscle (Irving, 2017), and in porcine muscle (Ma et al., 2021), they have not been consistently seen in rodent cardiac muscle under diastolic conditions (Ait-Mou et al., 2016; Caremani et al., 2019). In any event, the structural changes responsible for increases S_{M6} appear to disrupt the stabilizing interactions between OFF myosin heads and the thick filament backbone, most likely involving titin and MyBP-C (Dutta et al., 2023; Tamborrini et al., 2023).

7. Line 99-102. Sarcomere length-dependent mechanisms are not fully elucidated. In resting skeletal muscle, stretch increases thick filament stress and triggers the release of the myosin motors from their folded conformation by a mechano-sensing mechanism (5). In cardiac muscle, the picture is less clear. Some x-ray diffraction studies have shown that stretch does not affect the OFF state of the thick filament in intact cardiac muscle during diastole (6), while others have found structural changes in thick filaments (7).

We incorporated these details and citations into the paper.

From line 116:

“It is important to note that these ideas are actively evolving, and the evidence is more consistent in skeletal muscle (Irving, 2017), but in cardiac muscle, the picture is less clear, with some indicating stretch does not affect the OFF state of the thick filament in intact cardiac muscle during diastole (Caremani et al., 2019), while others have found structural changes in thick filaments (Ait-Mou et al., 2016).”

8. Line 125 and line 146, do the authors mean I1,1/I1,0?

Yes. We now corrected this and checked for this mistake throughout the paper.

9. Line 159, suggestion: by an unknown mechanism.

Changed as suggested.

10. Line 174, do the authors mean weak-binding crossbridges?

In this case, we mean force-producing crossbridges.

We clarified the statement (from line 199):

“In this study, C2^{-/-} sarcomeres have more ON myosin heads as well as longer thin filaments. In passive muscles, ON myosins produce a small number of force-producing crossbridges that generate a small amount of force on the thin filaments (Donaldson et al., 2012; Eakins et al., 2016; Jarvis et al., 2021; Regnier et al., 1995). These bound crossbridges, estimated at ~2% of myosin heads, increase with increasing proportion of ON myosin heads (Prodanovic et al., 2023). In C2^{-/-} fibers, more myosin heads are ON (see above), and so we hypothesize that more force-producing crossbridges are engaged as well, contributing to thin filament extension.”

11. Line 180-181. Suggested edit: It is not known how this pool of weak-binding myosin crossbridges and changes in thin filament length modulate muscle contraction.”

Changed as suggested, but with the term “force-producing crossbridges”.

1. T. Song et al., Fast skeletal myosin-binding protein-C regulates fast skeletal muscle contraction. Proc Natl Acad Sci U S A 118 (2021).
2. A. L. Hessel et al., Myosin-binding protein C forms C-links and stabilizes OFF states of myosin. bioRxiv 10.1101/2023.09.10.556972 (2023).
3. T. C. Irving, R. Craig, Getting into the thick (and thin) of it. J Gen Physiol 151, 610-613 (2019).
4. L. Fusi, E. Brunello, Z. Yan, M. Irving, Thick filament mechano-sensing is a calcium-independent regulatory mechanism in skeletal muscle. Nat Commun 7, 13281 (2016).
5. M. Irving, Regulation of Contraction by the Thick Filaments in Skeletal Muscle. Biophys J 113, 2579-2594 (2017).
6. M. Caremani et al., Inotropic interventions do not change the resting state of myosin motors during cardiac diastole. J Gen Physiol 151, 53-65 (2019).
7. Y. Ait-Mou et al., Titin strain contributes to the Frank-Starling law of the heart by structural rearrangements of both thin- and thick-filament proteins. Proc Natl Acad Sci U S A 113, 2306-2311 (2016).

Reviewer #3 (Remarks to the Author):

This is a nice study, building on the authors' expertise on this subject.

Summary

In this study, Dr Hessel and team have used the fast skeletal muscle from a mouse with fast-MyBP-C-ko, labelled C2^{+/-}, from the lab of Sadayappan. Fast skeletal fibres have both fast and slow MyBP-C, hence the C2^{-/-} muscles have only slow MyBP-C (~20% in EDL muscle). Here Hessel and team performed mechanical experiments and carried out X-ray analysis of demembrated EDL NTG and C2^{-/-} fibres at

two different sarcomere lengths, 2.4 and 2.8 μm . This is a good choice of SL as it is within the range where linear effects were observed in previous studies (eg Hessel et al, PNAS, 2022). They report reduced Ca sensitivity and reduced active tension for the ko fibres. X-ray analysis shows higher I11/10 values for the ko fibres, in agreement with previous studies. They examined periodicities of thick and thin filaments, observing increased values for both filaments NTG fibres but little change in the ko.

Comments

I found this a hard paper to review because of the lack of care in the writing in some places.

To talk about movement of mass from myosin to actin, we refer to the ratio I11/10 as in the axis label in Fig 2e. So why not use this consistently rather than refer to I10/11 as in the text on line 125 and line 145. I10/11 gives the inverse ratio which I presume was not the intention of the authors. On line 151 the ratio is stated correctly.

The reviewer is correct (I11/I10). We apologize for this oversight and fixed throughout.

What I found troublesome is the major error for the captions in Figure 1 and 2, that C2-/- is represented by black/squares and NTG by red/triangles. This is wrong and the reverse is correct. If the authors knew about it after submission, they should have immediately informed the editorial office so that the reviewers could be informed. In the key for Fig 1b and c, the correct colour convention is used, black for NTG and red for ko. I wasted a lot of time grappling with this.

We apologize for this oversight and wasting the reviewer's time and have fixed this error.

Discussion about X-ray evidence for thick and thin filament length change.

The authors have examined reflections originating from the thick and thin filaments and they report an increase in nm in the spacings of these reflections. On the other hand, it was reported long ago that upon activation, the thick filament undergoes an increase of $\sim 1\%$. I think it is misleading to talk about change in filament length per se as it implies a change in the actual length as in skeletal versus cardiac muscles (see Granzier et al, Am J Physiol, 1991). A small change like $<1\%$ can be brought about by changes in the molecular packing.

We agree that discussion on the filament length can be misleading. We clarify that the changes in periodicities indicate very small changes to the filament length, per se, but indicate much more meaningful changes to the orientation of the periodic structures and/or changes to filament packing.

From line 111:

“It is important to note that these ideas are actively evolving. While there is consensus that the spacing of the M6 reflection (S_{M6}) reflects the periodicity of structures within the thick filament backbone, where these small changes in thick filament backbone periodicity ($<1\%$) are due to actual extension of molecules themselves, or changes in the molecular packing, or both, has been a subject of considerable debate and is not yet clear (Dutta et al., 2023; Huxley et al., 2003; Zoghbi et al., 2008).”

From line 138:

“The spacing of the M6 reflection (S_{M6}) reflects changes within the thick filament periodicity length, either due to extension of individual molecules comprising the backbone or changes in the molecular packing of these molecules within the filament (Huxley et al., 2003).”

From line 184:

“By an unknown mechanism, thin filaments backbones change their conformation and/or subtly elongate with increasing SL in passive mammalian cardiac and skeletal muscle (Ait-Mou et al., 2016; Hessel et al., 2022; Hessel et al., 2023) and we evaluated if this changes in fMyBP-C KO muscle.”

I think the authors should also report the changes as a percentage change of the total length of the filaments.

As suggested by the reviewer, we also added a supplementary information figure (below) that provides the relative changes to these and all X-ray structures at 2.8 μm SL (normalized to the values at 2.4 μm SL).

Minor comments

In both figures, the black squares and red triangles are hard to see because of their fine hairline outlines. I suggest the authors use coloured filled-in circles for the points in the graph eg like Fig 4c in Song et al, Fast skeletal MyBP-C regulates fast skeletal muscle, PNAS, 2021. Please note that saturated colours in tiny graphs like Fig 4c of Song are much easier to see.

We updated the figure to fill in the squares and rectangles (shown below).

Line 67 Re fig 1 b and c

In figure legend, clarify which is the dashed line, by drawing a dashed line with two dashes

We updated the figure as requested (shown below).

Line 75 Fig 1f, SL-independent reduction in Ca sensitivity
 Fig 1h Put key in the figure for the colours: orange=2.4um

We updated the figure as requested (shown below).

Line 155-157 This is a very obscure sentence, please clarify.

The original sentence read: “Taken together, the fMyBP-C KO in C2^{-/-} present altered SL-independent and SL-dependent regulation of myosin head ON/OFF states that is likely linked to an inability to stabilize the OFF state”.

We updated the sentence to read (from line 180):

“Taken together, we find that compared to NTG, the measured alterations of myosin head regulation in the C2^{-/-} can be explained by a partial destabilization of the OFF state.”

Line 438 In the I-band, titin extends from the Z-disk to the tops of the ...
Surely it should be “tips” of the thick filament

This typo was corrected.

References

- Ackermann, M. A. and Kontrogianni-Konstantopoulos, A.** (2011). Myosin binding protein-C slow is a novel substrate for protein kinase A (PKA) and C (PKC) in skeletal muscle. *J. Proteome Res.* **10**, 4547–4555.
- Ackermann, M. A. and Kontrogianni-Konstantopoulos, A.** (2013). Myosin binding protein-C slow: a multifaceted family of proteins with a complex expression profile in fast and slow twitch skeletal muscles. *Front. Physiol.* **4**, 391.
- Adhikari, A. S., Trivedi, D. V., Sarkar, S. S., Song, D., Kooiker, K. B., Bernstein, D., Spudich, J. A. and Ruppel, K. M.** (2019). β -Cardiac myosin hypertrophic cardiomyopathy mutations release sequestered heads and increase enzymatic activity. *Nat. Commun.* **10**, 2685.
- Ait-Mou, Y., Hsu, K., Farman, G. P., Kumar, M., Greaser, M. L., Irving, T. C. and de Tombe, P. P.** (2016). Titin strain contributes to the Frank-Starling law of the heart by structural rearrangements of both thin- and thick-filament proteins. *Proc Natl Acad Sci USA* **113**, 2306–2311.
- Caremani, M., Pinzauti, F., Powers, J. D., Governali, S., Narayanan, T., Stienen, G. J. M., Reconditi, M., Linari, M., Lombardi, V. and Piazzesi, G.** (2019). Inotropic interventions do not change the resting state of myosin motors during cardiac diastole. *J. Gen. Physiol.* **151**, 53–65.
- de Tombe, P. P., Mateja, R. D., Tachampa, K., Ait Mou, Y., Farman, G. P. and Irving, T. C.** (2010). Myofilament length dependent activation. *J. Mol. Cell. Cardiol.* **48**, 851–858.
- Donaldson, C., Palmer, B. M., Zile, M., Maughan, D. W., Ikonomidis, J. S., Granzier, H., Meyer, M., VanBuren, P. and LeWinter, M. M.** (2012). Myosin cross-bridge dynamics in patients with hypertension and concentric left ventricular remodeling. *Circ. Heart Fail.* **5**, 803–811.
- Dutta, D., Nguyen, V., Campbell, K. S., Padrón, R. and Craig, R.** (2023). Cryo-EM structure of the human cardiac myosin filament. *Nature*.
- Eakins, F., Pinali, C., Gleeson, A., Knupp, C. and Squire, J. M.** (2016). X-ray Diffraction Evidence for Low Force Actin-Attached and Rigor-Like Cross-Bridges in the Contractile Cycle. *Biology (Basel)* **5**.
- Fusi, L., Brunello, E., Yan, Z. and Irving, M.** (2016). Thick filament mechano-sensing is a calcium-independent regulatory mechanism in skeletal muscle. *Nat. Commun.* **7**, 13281.
- Geist, J. and Kontrogianni-Konstantopoulos, A.** (2016). MYBPC1, an emerging myopathic gene: what we know and what we need to learn. *Front. Physiol.* **7**, 410.
- Gurnett, C. A., Desruisseau, D. M., McCall, K., Choi, R., Meyer, Z. I., Talerico, M., Miller, S. E., Ju, J.-S., Pestronk, A., Connolly, A. M., et al.** (2010). Myosin binding protein C1: a novel gene for autosomal dominant distal arthrogyrosis type 1. *Hum. Mol. Genet.* **19**, 1165–1173.

- Harris, S. P.** (2021). Making waves: A proposed new role for myosin-binding protein C in regulating oscillatory contractions in vertebrate striated muscle. *J. Gen. Physiol.* **153**.
- Hessel, A. L., Ma, W., Mazara, N., Rice, P. E., Nissen, D., Gong, H., Kuehn, M., Irving, T. and Linke, W. A.** (2022). Titin force in muscle cells alters lattice order, thick and thin filament protein formation. *Proc Natl Acad Sci USA* **119**, e2209441119.
- Hessel, A. L., Engels, N. M., Kuehn, M., Nissen, D., Sadler, R. L., Ma, W., Irving, T. C., Linke, W. A. and Harris, S. P.** (2023). Myosin-binding protein C forms C-links and stabilizes OFF states of myosin. *BioRxiv*.
- Hettige, P., Tahir, U., Nishikawa, K. C. and Gage, M. J.** (2020). Comparative analysis of the transcriptomes of EDL, psoas, and soleus muscles from mice. *BMC Genomics* **21**, 808.
- Huang, X., Torre, I., Chiappi, M., Yin, Z., Vydyanath, A., Cao, S., Raschdorf, O., Beeby, M., Quigley, B., de Tombe, P. P., et al.** (2023). Cryo-electron tomography of intact cardiac muscle reveals myosin binding protein-C linking myosin and actin filaments. *J. Muscle Res. Cell Motil.*
- Huxley, H. E., Reconditi, M., Stewart, A. and Irving, T.** (2003). X-ray interference evidence concerning the range of crossbridge movement, and backbone contributions to the meridional pattern. *Adv. Exp. Med. Biol.* **538**, 233–41; discussion 241.
- Irving, M.** (2017). Regulation of Contraction by the Thick Filaments in Skeletal Muscle. *Biophys. J.* **113**, 2579–2594.
- Irving, T. C. and Craig, R.** (2019). Getting into the thick (and thin) of it. *J. Gen. Physiol.* **151**, 610–613.
- Jarvis, K. J., Bell, K. M., Loya, A. K., Swank, D. M. and Walcott, S.** (2021). Force-velocity and tension transient measurements from *Drosophila* jump muscle reveal the necessity of both weakly-bound cross-bridges and series elasticity in models of muscle contraction. *Arch. Biochem. Biophys.* **701**, 108809.
- Linari, M., Brunello, E., Reconditi, M., Fusi, L., Caremani, M., Narayanan, T., Piazzesi, G., Lombardi, V. and Irving, M.** (2015). Force generation by skeletal muscle is controlled by mechanosensing in myosin filaments. *Nature* **528**, 276–279.
- Li, A., Nelson, S. R., Rahmanseresht, S., Braet, F., Cornachione, A. S., Previs, S. B., O’Leary, T. S., McNamara, J. W., Rassier, D. E., Sadayappan, S., et al.** (2019). Skeletal MyBP-C isoforms tune the molecular contractility of divergent skeletal muscle systems. *Proc Natl Acad Sci USA* **116**, 21882–21892.
- Ma, W., Gong, H., Kiss, B., Lee, E.-J., Granzier, H. and Irving, T.** (2018). Thick-Filament Extensibility in Intact Skeletal Muscle. *Biophys. J.* **115**, 1580–1588.
- Ma, W., Childers, M., Murray, J., Moussavi-Harami, F., Gong, H., Weiss, R., Daggett, V., Irving, T. and Regnier, M.** (2020). Myosin dynamics during relaxation in mouse soleus muscle and modulation by 2’-deoxy-ATP. *J Physiol (Lond)* **598**, 5165–5182.
- Ma, W., Henze, M., Anderson, R. L., Gong, H., Wong, F. L., Del Rio, C. L. and Irving, T.** (2021). The Super-Relaxed State and Length Dependent Activation in Porcine Myocardium. *Circ. Res.* **129**, 617–630.

- Ma, W., McMillen, T. S., Childers, M. C., Gong, H., Regnier, M. and Irving, T. (2023a).** Structural OFF/ON transitions of myosin in relaxed porcine myocardium predict calcium-activated force. *Proc Natl Acad Sci USA* **120**, e2207615120.
- Ma, W., Qi, L., Prodanovic, M., Gong, H. M., Zambataro, C., Gollapudi, S. K., Mijailovich, S. M., Del Rio, C. L., Nag, S. and Irving, T. C. (2023b).** Myosin in autoinhibited off state(s), stabilized by mavacamten, can be recruited via inotropic effectors. *Biophys. J.* **122**, 122a.
- Prodanovic, M., Wang, Y., Mijailovich, S. M. and Irving, T. (2023).** Using Multiscale Simulations as a Tool to Interpret Equatorial X-ray Fiber Diffraction Patterns from Skeletal Muscle. *Int. J. Mol. Sci.* **24**,.
- Reconditi, M. (2006).** Recent improvements in small angle x-ray diffraction for the study of muscle physiology. *Rep. Prog. Phys.* **69**, 2709–2759.
- Regnier, M., Morris, C. and Homsher, E. (1995).** Regulation of the cross-bridge transition from a weakly to strongly bound state in skinned rabbit muscle fibers. *Am. J. Physiol.* **269**, C1532-9.
- Sarkar, S. S., Trivedi, D. V., Morck, M. M., Adhikari, A. S., Pasha, S. N., Ruppel, K. M. and Spudich, J. A. (2020).** The hypertrophic cardiomyopathy mutations R403Q and R663H increase the number of myosin heads available to interact with actin. *Sci. Adv.* **6**, eaax0069.
- Schiaffino, S., Rossi, A. C., Smerdu, V., Leinwand, L. A. and Reggiani, C. (2015).** Developmental myosins: expression patterns and functional significance. *Skelet. Muscle* **5**, 22.
- Sitbon, Y. H., Yadav, S., Kazmierczak, K. and Szczesna-Cordary, D. (2020).** Insights into myosin regulatory and essential light chains: a focus on their roles in cardiac and skeletal muscle function, development and disease. *J. Muscle Res. Cell Motil.* **41**, 313–327.
- Song, T., McNamara, J. W., Ma, W., Landim-Vieira, M., Lee, K. H., Martin, L. A., Heiny, J. A., Lorenz, J. N., Craig, R., Pinto, J. R., et al. (2021).** Fast skeletal myosin-binding protein-C regulates fast skeletal muscle contraction. *Proc Natl Acad Sci USA* **118**,.
- Song, T., Landim-Vieira, M., Ozdemir, M., Gott, C., Kanisicak, O., Pinto, J. R. and Sadayappan, S. (2023).** Etiology of genetic muscle disorders induced by mutations in fast and slow skeletal MyBP-C paralogs. *Exp. Mol. Med.* **55**, 502–509.
- Tamborrini, D., Wang, Z., Wagner, T., Tacke, S., Stabrin, M., Grange, M., Kho, A. L., Rees, M., Bennett, P., Gautel, M., et al. (2023).** Structure of the native myosin filament in the relaxed cardiac sarcomere. *Nature* **623**, 863–871.
- Zoghbi, M. E., Woodhead, J. L., Moss, R. L. and Craig, R. (2008).** Three-dimensional structure of vertebrate cardiac muscle myosin filaments. *Proc Natl Acad Sci USA* **105**, 2386–2390.

Reviewers' comments:

Reviewer #1 (Remarks to the Author):

The authors have addressed all of my queries and suggestions. Based on the revisions made, I recommend the manuscript for publication. I look forward to witnessing the future scientific contributions of the authors to the field.

Reviewer #2 (Remarks to the Author):

The authors have successfully addressed most of my concerns. However, one major point remains. I asked about the 26% reduction in MLC2 expression in EDL muscle of C2-/- mice reported by Song et al. 2021 (SI Appendix, Fig. S13). I do not agree with the explanation provided by the authors that the 35% (?) reduction is in Myl2, which is a ventricular muscle isoform. They further claim that there is no change in the expression level of the predominant fast skeletal regulatory light chain isoform Mylpf. I believe that there is some nomenclature confusion. Mylpf is also known as MLC2. Song et al. 2021 validated the expression changes of the predominant isoform in EDL muscle (SI Appendix, Fig. S13). In their western blot, they used the polyclonal antibody against MYLPF (A24975). Song et al. make a point (albeit a very short one) about that in their discussion: "We also determined that total MLC2 protein is down-regulated in C2-/- EDL muscles, most likely contributing to the observed reduction in force and speed of contraction". It will be disingenuous to claim that there are no changes in the regulatory light chain. The authors need to fix that and write a few sentences in the discussion to explain how this is expected to affect their results/interpretation.

Minor comment. The concept of force-producing cross-bridges in resting muscle is not widely accepted because their number is difficult to measure precisely. It will require the use of a specific myosin inhibitor and some modelling, which the authors plan to do in the future to my understanding. The authors may want to use more careful wording (may, could etc.) but this is not a required edit just opinion/advice.

Reviewer #3 (Remarks to the Author):

I am satisfied with the response by the authors in the revised version to my comments for the first version.

Below we provide the original decision letter (reviewer comments portion only) and our replies in a red font. Changes to the text that are reproduced here are indicated by a blue font. Within the manuscript, changes are shown in red. In the rebuttal letter only, we show (name, year) citation style, so readers can quickly identify the citation.

Reviewers' comments:

Reviewer #1 (Remarks to the Author):

The authors have addressed all of my queries and suggestions. Based on the revisions made, I recommend the manuscript for publication. I look forward to witnessing the future scientific contributions of the authors to the field.

We thank the reviewer for their efforts on our manuscript. We are also excited to continue this line of research!

Reviewer #2 (Remarks to the Author):

The authors have successfully addressed most of my concerns. However, one major point remains. I asked about the 26% reduction in MLC2 expression in EDL muscle of C2^{-/-} mice reported by Song et al. 2021 (SI Appendix, Fig. S13). I do not agree with the explanation provided by the authors that the 35% (?) reduction is in Myl2, which is a ventricular muscle isoform. They further claim that there is no change in the expression level of the predominant fast skeletal regulatory light chain isoform Mylpf.

I believe that there is some nomenclature confusion. Mylpf is also known as MLC2. Song et al. 2021 validated the expression changes of the predominant isoform in EDL muscle (SI Appendix, Fig. S13). In their western blot, they used the polyclonal antibody against MYLFP (A24975). Song et al. make a point (albeit a very short one) about that in their discussion: “We also determined that total MLC2 protein is down-regulated in C2^{-/-} EDL muscles, most likely contributing to the observed reduction in force and speed of contraction”. It will be disingenuous to claim that there are no changes in the regulatory light chain. The authors need to fix that and write a few sentences in the discussion to explain how this is expected to affect their results/interpretation.

The reviewer is correct to surmise there was some confusion with the nomenclature. We thank the reviewer for pointing out this issue. The antibody used in Song et al. 2021 was indeed against Mylpf, the regulatory light chain that accompanies myosin IIb (Myh4) in mouse EDL. As noted by the reviewer, Song et al. 2021 report a ~25% reduction in Mylpf detected by Western blot (Figure S13). They also report trends for a ~15% reduction in Myh4 and myomesin-1 (Myom1) by Western blot that were not statistically significant (Figure S14).

We have asked our coauthors, Song and Sadayappan, to clarify the results of the mass spectrometry, which we would accept to be more quantitative compared to Western blot. They report significant reductions in Myh4 by ~18% and Myom1 by ~12% using mass spectrometry (Table S8). They did not find a change in Mylpf content by mass spectrometry (unpublished). Nevertheless, collectively, these data (mass spect and Western blots) suggest that the total myofilament content is reduced in the EDL of the C2^{-/-} mouse on the order of ~20% as detected for different proteins and modalities. This reduction in total myofilament protein would certainly contribute to the ~40% reduction in in-vivo grip strength and ex-vivo maximum force production but cannot account for the reduction in length-dependent activation we report with mechanics and X-ray data in the current work. We note this qualification in the manuscript in the Introduction.

From line 54:

“The NTG EDL is predominately a fast-twitch muscle with a ~43% fMyBP-C expression and ~57% sMyBP-C (Li et al., 2019). $C2^{-/-}$ express trace levels of fMyBP-C with a ~15% increase in sMyBP-C that leads to thick filaments with ~33% reduction in total MyBP-C molecules (Song et al., 2021). This model also presents a reduction in myosin heavy chain (Myh4) by 18%, myosin regulatory light chain fast isoform (MyIpf) by 25%, myomesin-1 (Myom1) by 12%, which collectively suggests a reduced myofilament protein content on the order of 20% (Song et al., 2021). This loss of the force-producing apparatus could contribute to the observed reduction in maximum force production observed in the EDL of these mice²⁰, but it cannot account for many functional and structural changes we find.”

Minor comment. The concept of force-producing cross-bridges in resting muscle is not widely accepted because their number is difficult to measure precisely. It will require the use of a specific myosin inhibitor and some modelling, which the authors plan to do in the future to my understanding. The authors may want to use more careful wording (may, could etc.) but this is not a required edit just opinion/advice.

We appreciate this comment. We now tone down the wording on this point.

From line 200:

“In this study, $C2^{-/-}$ sarcomeres have more ON myosin heads as well as longer thin filaments. In passive muscles, ON myosins likely produce a small number of force-producing crossbridges that generate a small amount of force on the thin filaments (Donaldson et al., 2012; Eakins et al., 2016; Jarvis et al., 2021; Regnier et al., 1995). These bound crossbridges, estimated at ~2% of myosin heads, increase with increasing proportion of ON myosin heads (Prodanovic et al., 2023). In $C2^{-/-}$ fibers, more myosin heads are ON (see above), and so we hypothesize that more force-producing crossbridges are engaged as well, contributing to thin filament extension. This hypothesis can be tested by using mavacamten or dATP to force nearly all myosin heads into the OFF or ON state, respectively (Ma et al., 2023a; Ma et al., 2023b), and tracking changes to thin filament length. It is not known how this increased pool of force-producing myosin crossbridges and changes in thin filament length modulate muscle contraction.”

Reviewer #3 (Remarks to the Author):

I am satisfied with the response by the authors in the revised version to my comments for the first version.

We thank the reviewer for their efforts on our manuscript.

References

- Donaldson, C., Palmer, B. M., Zile, M., Maughan, D. W., Ikonomidis, J. S., Granzier, H., Meyer, M., VanBuren, P. and LeWinter, M. M. (2012). Myosin cross-bridge dynamics in patients with hypertension and concentric left ventricular remodeling. *Circ. Heart Fail.* **5**, 803–811.
- Eakins, F., Pinali, C., Gleeson, A., Knupp, C. and Squire, J. M. (2016). X-ray Diffraction Evidence for Low Force Actin-Attached and Rigor-Like Cross-Bridges in the Contractile Cycle. *Biology (Basel)* **5**.
- Jarvis, K. J., Bell, K. M., Loya, A. K., Swank, D. M. and Walcott, S. (2021). Force-velocity and tension transient measurements from *Drosophila* jump muscle reveal the necessity of both weakly-bound cross-bridges and series elasticity in models of muscle contraction. *Arch. Biochem. Biophys.* **701**, 108809.
- Li, A., Nelson, S. R., Rahmanseresht, S., Braet, F., Cornachione, A. S., Previs, S. B., O’Leary, T. S., McNamara, J. W., Rassier, D. E., Sadayappan, S., et al. (2019). Skeletal MyBP-C isoforms tune the

molecular contractility of divergent skeletal muscle systems. *Proc Natl Acad Sci USA* **116**, 21882–21892.

Ma, W., McMillen, T. S., Childers, M. C., Gong, H., Regnier, M. and Irving, T. (2023a). Structural OFF/ON transitions of myosin in relaxed porcine myocardium predict calcium-activated force. *Proc Natl Acad Sci USA* **120**, e2207615120.

Ma, W., Qi, L., Prodanovic, M., Gong, H. M., Zambataro, C., Gollapudi, S. K., Mijailovich, S. M., Del Rio, C. L., Nag, S. and Irving, T. C. (2023b). Myosin in autoinhibited off state(s), stabilized by mavacamten, can be recruited via inotropic effectors. *Biophys. J.* **122**, 122a.

Prodanovic, M., Wang, Y., Mijailovich, S. M. and Irving, T. (2023). Using Multiscale Simulations as a Tool to Interpret Equatorial X-ray Fiber Diffraction Patterns from Skeletal Muscle. *Int. J. Mol. Sci.* **24**,.

Regnier, M., Morris, C. and Homsher, E. (1995). Regulation of the cross-bridge transition from a weakly to strongly bound state in skinned rabbit muscle fibers. *Am. J. Physiol.* **269**, C1532-9.

Song, T., McNamara, J. W., Ma, W., Landim-Vieira, M., Lee, K. H., Martin, L. A., Heiny, J. A., Lorenz, J. N., Craig, R., Pinto, J. R., et al. (2021). Fast skeletal myosin-binding protein-C regulates fast skeletal muscle contraction. *Proc Natl Acad Sci USA* **118**,.

REVIEWERS' COMMENTS:

Reviewer #2 (Remarks to the Author):

The authors have addressed all my concerns and I recommend the article for publication.